# Ion pair sites for efficient electrochemical extraction of uranium in real nuclear wastewater

Tao Lin[1], Tao Chen[1], Chi Jiao[2], Haoyu Zhang[1], Kai Hou[1], Hongxiang Jin[1], Yan Liu ![ORCID][2] ✉, Wenkun Zhu ![ORCID][1] ✉ & Rong He ![ORCID][1] ✉

Electrochemical uranium extraction from nuclear wastewater represents an emerging strategy for recycling uranium resources. However, in nuclear fuel production which generates the majority of uranium-containing nuclear wastewater, fluoride ion ($F^-$) co-exists with uranyl ($UO_2^{2+}$), resulting in the complex species of $UO_2F_x$ and thus decreasing extraction efficiency. Herein, we construct $Ti^{\delta+}$-$PO_4^{3-}$ ion pair extraction sites in $Ti(OH)PO_4$ for efficient electrochemical uranium extraction in wastewater from nuclear fuel production. These sites selectively bind with $UO_2F_x$ through the combined Ti-F and multiple O-U-O bonds. In the uranium extraction, the uranium species undergo a crystalline transition from $U_3O_7$ to $K_3UO_2F_5$. In real nuclear wastewater, the uranium is electrochemically extracted with a high efficiency of 99.6% and finally purified as uranium oxide powder, corresponding to an extraction capacity of 6829 mg g$^{-1}$ without saturation. This work paves an efficient way for electrochemical uranium recycling in real wastewater of nuclear production.

Uranium is the key fundamental resource in the nuclear industry[1–3]. With the development of the nuclear industry, uranium resources in terrestrial ore will be depleted within less than a century, accompanied by the massive generation of uranium-containing nuclear wastewater[4,5]. In practical situations, the majority of uranium-containing nuclear wastewater is produced by the procedures of fuel production, such as uranium enrichment, uranium conversion, and fuel element fabrication, which requires the wide usage of uranium fluoride[6]. After the hydrolysis of uranium fluoride, the uranium in nuclear wastewater commonly exists in the form of uranyl ($UO_2^{2+}$), together with the co-existing high concentration of fluoride ion ($F^-$)[7]. As such, extracting uranium under the interference of high concentrations of $F^-$ is an important issue for environmental protection and the recycling of uranium resources[8–10]. For this nuclear wastewater, the traditional adsorption or ion exchange method commonly requires the pre-precipitation of $F^-$ by $Ca^{2+}$, resulting in the formation of uranium-containing $CaF_2$ as radioactive solid waste[11,12]. Accordingly,

searching for other technologies for uranium extraction is highly desired under the interference of high concentrations of $F^-$.

As an emerging technology, electrochemical uranium extraction has attracted ever-increasing attention due to the fast kinetics, increased extraction capacity, and resistance to the interference of anions[13–15]. At present, electrochemical uranium extraction has been applied in the aqueous systems without $F^-$[16,17]. For example, the Fe-$N_x$-C single atoms with functional amidoxime groups[18] were reported to achieve an extraction capacity of 1.2 mg g$^{-1}$ from real seawater over 24 h. However, despite the significant progress, the electrochemical uranium extraction in real nuclear wastewater with the existence of $F^-$ is still challenging, because of the rather complex species of uranium due to the coordination effect[19,20]. Specifically, the dominant hexavalent uranium (U(VI)) species generally lies on a series of uranyl fluoride ($UO_2F_x$)[21], such as $UO_2F^+$, $UO_2F_2$, and $UO_2F_3^-$, instead of bare $UO_2^{2+}$. In this case, the conventional uranium extraction sites suffer from the competing coordination of uranium and $F^-$, thus resulting in the poor extraction efficiency

[1]State Key Laboratory of Environment-friendly Energy Materials, School of National Defense, School of Life Science & Engineering, School of Materials & Chemistry, National Co-innovation Center for Nuclear Waste Disposal & Environmental Safety, Sichuan Civil-military Integration Institute, Southwest University of Science & Technology, Mianyang, P. R. China. [2]School of Chemistry and Materials Science, Anhui Normal University, Wuhu, P. R. China. ✉e-mail: ly0201@ahnu.edu.cn; zhuwenkun@swust.edu.cn; her@swust.edu.cn

of uranium under the interference of high concentrations of F⁻. Therefore, the construction of specific sites for the binding of $UO_2F_x$ is an essential prerequisite for achieving efficient electrochemical uranium extraction in real nuclear wastewater with high concentrations of F⁻.

Herein, we develop the $Ti(OH)PO_4$ with $Ti^{\delta+}$-$PO_4^{3-}$ ion pair extraction sites for selective binding of $UO_2F_x$ and efficient electrochemical uranium extraction under the interference of high concentration of F⁻. Both synchrotron X-ray absorption fine structure (XAFS) and theoretical simulation demonstrate that the $Ti^{\delta+}$-$PO_4^{3-}$ ion pair extraction sites are strongly bound with $UO_2F_x$ through the combined interactions of $Ti^{\delta+}$-F⁻ and $PO_4^{3-}$-$UO_2^{2+}$. During the uranium extraction, the $UO_2F_x$ is reduced to $U_3O_7$ as a gray deposit followed by further oxidation and crystallization as $K_3UO_2F_5$. In 400 mL of real nuclear wastewater produced by fuel production, the $Ti(OH)PO_4$ achieves an electrochemical extraction efficiency of 99.6% for uranium within 7 h, with an extraction capacity of 6829 mg g⁻¹ without saturation. The uranium-containing powder is successfully collected from the real nuclear wastewater, with 92.1% of uranium proportion among the metal elements in the powder.

## Results

### Construction of Ti(OH)PO₄ with ion pair sites

The construction of ion pair sites originated from the reconstruction of lamellar OH-terminated $Ti_3C_2$, as illustrated in Fig. 1a. Initially, the Al layers of commercial bulk $Ti_3AlC_2$ were etched to form the accordion-like stack of F-terminated $Ti_3C_2$ nanosheets. Followed by the exfoliation process under $N_2$ protection in aqueous solution, the stacked F-terminated $Ti_3C_2$ nanosheets were transformed into lamellar OH-terminated $Ti_3C_2$ nanosheets (Supplementary Fig. 1). The flexible exfoliated lamellar OH-terminated $Ti_3C_2$ nanosheets were confirmed by the high-resolution transmission electron microscope (HRTEM) images, and the X-ray diffraction (XRD) pattern[22–24] (Supplementary Fig. 2). As shown by Fourier transform infrared spectroscopy (FT-IR) measurement, the exfoliated lamellar OH-terminated $Ti_3C_2$ nanosheets lacked the peak of C-F stretching at 1080 cm⁻¹ relative to accordion-like stacked $Ti_3C_2$ nanosheets, demonstrating the replacement of -F terminations by -OH terminations during the exfoliation process (Supplementary Fig. 3). After that, a wet chemical treatment was applied to transform the exfoliated lamellar OH-terminated $Ti_3C_2$ into $Ti(OH)PO_4$, which contained an ion pair of $Ti^{\delta+}$ ($\delta < 4$) and $PO_4^{3-}$. As shown by the transmission electron microscopy (TEM) images, the $Ti(OH)PO_4$ exhibited a nanorod-like morphology (Fig. 1b). The atomic force microscopy (AFM) analysis and the corresponding three-dimensional graph of $Ti(OH)PO_4$ nanorods indicated the narrow height distribution due to the regular stacking (Supplementary Fig. 4). Figure 1c showed the HRTEM image of a free-standing $Ti(OH)PO_4$ nanocrystal. The $Ti(OH)PO_4$ nanorod exhibited a periodic layer-by-layer structure with a

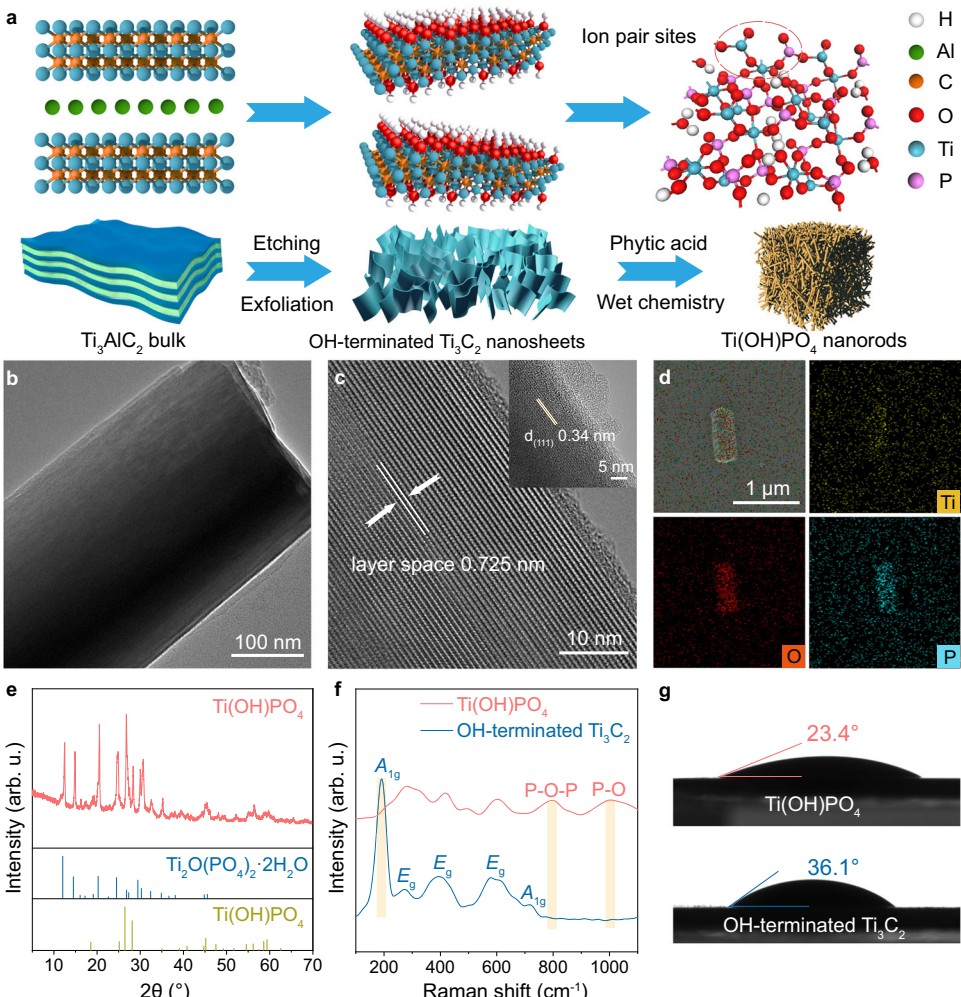

**Fig. 1 | Construction of ion pair sites. a** The schematic illustration for the preparation of the $Ti(OH)PO_4$ with ion pair sites. **b** TEM image of $Ti(OH)PO_4$ nanorods. **c** HRTEM image of $Ti(OH)PO_4$ nanorods. Inset: HRTEM image of the boundary of $Ti(OH)PO_4$ nanorods. **d** EDS analysis of $Ti(OH)PO_4$ nanorods. **e** XRD pattern, **f** Raman spectra, and **g** static contact angle of $Ti(OH)PO_4$ nanorods and lamellar OH-terminated $Ti_3C_2$ nanosheets. The highlighted areas in Fig. 1f show the $A_{1g}$ peak of OH-terminated $Ti_3C_2$ and the P-O-P and P-O peaks of $Ti(OH)PO_4$, respectively. Credits: **a** (schematic models) copyright Hangzhou SPHERE Technology Co., Ltd. Source data are provided as a Source Data file.

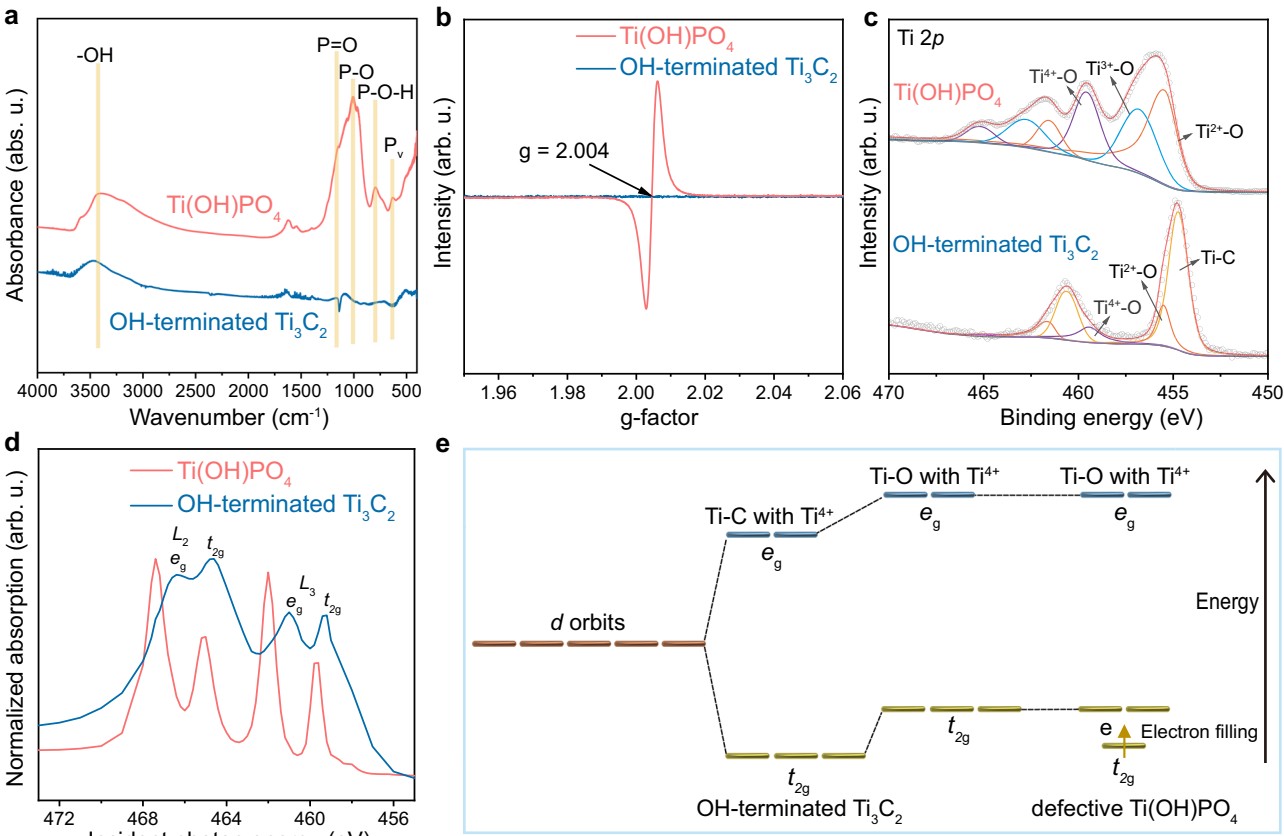

**Fig. 2 | Bonding and electronic structure of ion pair sites. a** FT-IR spectrum, **b** ESR spectrum, **c** Ti 2$p$ XPS spectrum, and **d** XANES spectrum of Ti(OH)PO$_4$ nanorods and lamellar OH-terminated Ti$_3$C$_2$ nanosheets. **e** Energy diagram of the splitting orbits of Ti 3$d$ peak. $L_2$ and $L_3$ represent two different sub-energy levels in the $L$ shell of the Ti atom, respectively. $e_g$ and $t_{2g}$ represent the energy level splitting of $d$ orbitals in the coordination field. Source data are provided as a Source Data file.

spacing of 0.725 nm, which was larger than the diameter of uranyl or uranyl fluoride (<0.4 nm). The inset was the edge region of the rod structure, which displayed the lattice fringe with interlayer spacing of 0.34 nm and indicated the (111) facets of Ti(OH)PO$_4$. Determined by energy dispersive X-ray spectroscopy (EDS), the Ti, O, and P elements filled the outline of the selected region, indicating the uniform distribution of the elemental components in the Ti(OH)PO$_4$ nanorod (Fig. 1d).

The physical phase of Ti(OH)PO$_4$ nanorods was further validated. Two sets of diffraction peaks appeared in the XRD pattern of the Ti(OH)PO$_4$ nanorods (Fig. 1e). The two sets of characteristic peaks were identified as the standard monoclinal Ti(OH)PO$_4$ phase (JCPDS #36-0697) and Ti$_2$O(PO$_4$)$_2$·2H$_2$O (JCPDS #52-1529). Notably, the Ti$_2$O(PO$_4$)$_2$·2H$_2$O was the dimerization of Ti(OH)PO$_4$ after a dehydration process, indicating the transformable hydroxyl in Ti(OH)PO$_4$. Figure 1f shows the Raman spectra of Ti(OH)PO$_4$ and lamellar OH-terminated Ti$_3$C$_2$. After the reconstruction of OH-terminated Ti$_3$C$_2$ into Ti(OH)PO$_4$, the $A_{1g}$ peak of pristine OH-terminated Ti$_3$C$_2$ disappeared, accompanied by the characteristic peaks of Ti-O bonds and P-O bonds[25,26]. Moreover, the Ti(OH)PO$_4$ exhibited a water static contact angle of 23.4°, which was lower than that of pristine lamellar OH-terminated Ti$_3$C$_2$ (Fig. 1g). Such phenomenon resulted from the incorporation of hydroxyl into lattice with the enhanced hydrogen bonds in Ti(OH)PO$_4$, further demonstrating the existence form of Ti(OH)PO$_4$.

The ion pair sites in Ti(OH)PO$_4$ were investigated by the analysis of the bonding and electronic structure of Ti(OH)PO$_4$. Determined by FT-IR measurement (Fig. 2a), the stretching vibrations of P=O, P-O, and P-O-H in PO$_4^{3-}$ group were observed in the spectrum of Ti(OH)PO$_4$, which differed from that of OH-terminated Ti$_3$C$_2$[27]. In the electron spin

resonance (ESR) spectra (Fig. 2b), the Ti(OH)PO$_4$ exhibited a sharp resonance signal at the g-factor of 2.004, indicating the massive oxygen vacancies in Ti(OH)PO$_4$[28]. Moreover, the X-ray photoelectron spectroscopy (XPS) analysis was performed to check the chemical bonds of Ti(OH)PO$_4$, which displayed the Ti, P, and O signals in the survey spectra (Supplementary Fig. 5a). The oxidative state of P in the PO$_4^{3-}$ group was confirmed by the P 2$p$ spectrum of Ti(OH)PO$_4$ (Supplementary Fig. 5b). In the O 1$s$ spectrum, the peak of the O in the vicinity of oxygen vacancy was observed at 531.2 eV, further demonstrating the defective nature of Ti(OH)PO$_4$[29] (Supplementary Fig. 5c). Meanwhile, Ti$_3$C$_2$ possessed the dominant signal of Ti-C bond, together with the slight signal of Ti$^{4+}$-O and Ti$^{2+}$-O bonds, which was attributed to the -OH termination (Fig. 2c). By comparison, the Ti 2$p$ spectrum of Ti(OH)PO$_4$ showed three sets of peaks, which were assigned as Ti$^{4+}$-O, Ti$^{3+}$-O, and Ti$^{2+}$-O[30–32]. The massive low valent Ti$^{\delta+}$ ($\delta$ < 4) indicated the defective nature of Ti(OH)PO$_4$, which was constructed by the oxygen vacancies during the transformation of OH-terminated Ti$_3$C$_2$ into Ti(OH)PO$_4$.

To further demonstrate the electronic structure of Ti in Ti(OH)PO$_4$ nanorods, we tested the $L$-edge X-ray absorption near-edge structure (XANES) spectra of hydroxyl-terminated Ti$_3$C$_2$ and Ti(OH)PO$_4$. As shown in Fig. 2d, both the $L_2$ edge and $L_3$ edge of Ti(OH)PO$_4$ nanorods exhibited a slight shift of 2$p$ to $t_{2g}$ transition and a dramatic shift of 2$p$ to $e_g$ transition relative to pristine OH-terminated Ti$_3$C$_2$. This result was illustrated by the energy diagram of the splitting orbits of the Ti 3$d$ peak[33] (Fig. 2e). Typically, Ti possessed five 3$d$ orbits, which could be divided into two $e_g$ orbits and three $t_{2g}$ orbits[34,35]. The binding energy of $e_g$ and $t_{2g}$ orbits was associated with two factors, including the coordination environment and the electron filling of Ti. For the coordination environment, the transformation of Ti-C in

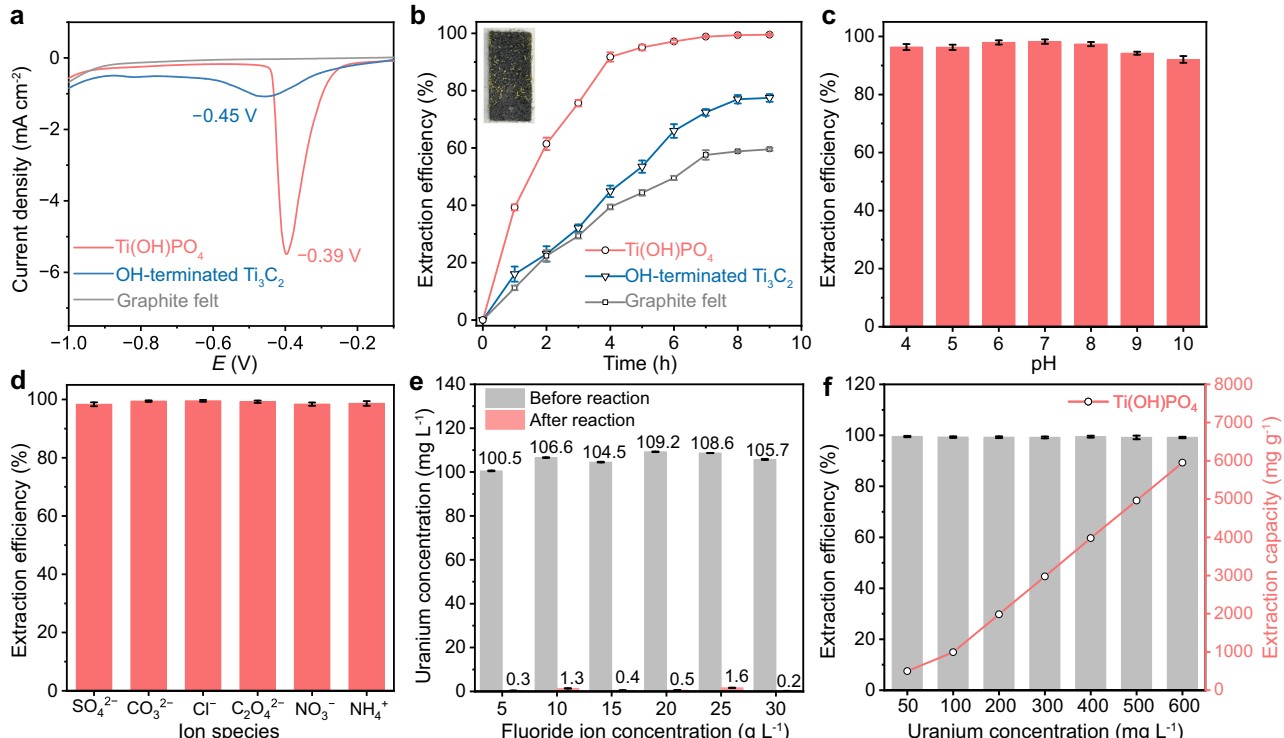

**Fig. 3 | Electrochemical uranium extraction of ion pair sites. a** LSV tests on Ti(OH)PO$_4$, OH-terminated Ti$_3$C$_2$, and bare graphite felt electrodes. $E$ represents potential. **b** The electrochemical extraction efficiency of U(VI) on Ti(OH)PO$_4$, OH-terminated Ti$_3$C$_2$, and bare graphite felt electrodes. Inset: macroscopic deposition stripes of crystalline uranium. The stability of electrochemical extraction of U(VI) on Ti(OH)PO$_4$ electrode **c** under different pH conditions, **d** with different co-existing ionic species. **e** The change in uranium concentration after electrochemical uranium extraction under electrolytes with different F$^-$ concentrations (ranging from 5 g L$^{-1}$ to 30 g L$^{-1}$). **f** The electrochemical extraction efficiency of U(VI) on Ti(OH)PO$_4$ as the initial concentration of U(VI) varied from 50 mg L$^{-1}$ to 600 mg L$^{-1}$. Error bars represent standard deviation of three measurements. Source data are provided as a Source Data file.

OH-terminated T$_3$C$_2$ into Ti-O in Ti(OH)PO$_4$ induced the positive shift of both $e_g$ and $t_{2g}$ orbits in Ti $L$-edge XANES, owing to the higher electronegativity of O than that of C. Moreover, for the factor of electron filling, Ti$^{4+}$ possessed empty $e_g$ and $t_{2g}$ orbits, whereas defective Ti(OH)PO$_4$ possessed abundant Ti$^{3+}$ and Ti$^{2+}$, which respectively had one and two electrons in 3$d$ orbits. Due to the lower energy of $t_{2g}$ orbits than $e_g$ orbits, the additional electrons in Ti$^{\delta+}$ were located at $t_{2g}$ orbits, resulting in the dramatic energy decrease of $t_{2g}$ orbits. As a comprehensive result of the coordination environment and the electron filling of Ti, the $e_g$ peak in Ti(OH)PO$_4$ showed an obvious shift, whereas $t_{2g}$ peak displayed a negligible shift relative to OH-terminated T$_3$C$_2$ in Ti $L$-edge XANES (Supplementary Table 1). The Ti$^{\delta+}$ and the PO$_4^{3-}$ group in Ti(OH)PO$_4$ constructed the ion pair sites, which were the key species for the uranium extraction under the interference of high concentration of F$^-$.

### Electrochemical uranium extraction

The Ti(OH)PO$_4$ nanorods were uniformly spread on graphite felt to evaluate the effect of ion pair sites on the electrochemical uranium extraction. An aqueous solution containing 100 mg L$^{-1}$ of U(VI) and 30 g L$^{-1}$ of F$^-$ was adopted as the electrolyte to simulate the wastewater of nuclear production. We demonstrated the reduction peak of U(VI) in this electrolyte by linear sweep voltammetry (LSV) test (Fig. 3a). The Ti(OH)PO$_4$ nanorods exhibited a sharp reduction peak at −0.39 V vs. Ag/AgCl, which corresponded to the reduction of U(VI) into pentavalent uranium (U(V)). By comparison, these peaks of pristine OH-terminated Ti$_3$C$_2$ and the bare graphite felt were located at more negative positions with rather weak intensity. To double-check the origin of the reduction peak, we performed Cyclic Voltammetry (CV) tests in the presence or absence of U(VI) in 30 g L$^{-1}$ of F$^-$ (Supplementary Fig. 6). The overall current density of the CV curve

dramatically increased after the addition of U(VI), indicating that the U(VI) participated in the composition of double-layer capacitance[36]. Besides, the reduction peak disappeared in the absence of U(VI), further demonstrating the ascription of the uranium reduction peak. These results indicated the Ti(OH)PO$_4$ nanorods possessed a special activity for uranium extraction in the presence of F$^-$.

As shown in Fig. 3b, the electrochemical extraction efficiency of U(VI) on Ti(OH)PO$_4$ was 95.1% within 5 h, which was significantly superior to that of OH-terminated Ti$_3$C$_2$ (53.5%) and that of bare graphite felt (44.4%). Further discussion on the termination group of Ti$_3$C$_2$ indicated the -OH group was the extraction site, instead of -F (Supplementary Fig. 7 and Supplementary Table 2). Notably, the ultimate extraction efficiency of U(VI) from Ti(OH)PO$_4$ attained 99.5%, which enabled the direct observation of macroscopic deposition stripes of crystalline uranium (inset of Fig. 3b). In addition, compared with the non-voltage adsorption method, the electrochemical extraction efficiency of U(VI) on Ti(OH)PO$_4$ nanorods exhibited a 2-fold enhancement (Supplementary Fig. 8). To collect the uranium from the electrode, 0.1 mol of HCl solution was taken as the desorbing agent. After 10 seconds of washing time, the yellow stripe-like crystallization on the surface of the electrode gradually dispersed in the desorbing solution (Supplementary Fig. 9). As a result, the electrochemical method with Ti(OH)PO$_4$ as an electrode represented an efficient protocol for uranium extraction in the presence of a high concentration of F$^-$.

The stability of electrochemical extraction of U(VI) on the Ti(OH)PO$_4$ nanorods was further evaluated. After 5 cycles of uranium extraction, the extraction efficiency of U(VI) on Ti(OH)PO$_4$ retained a high value of 92.2% (Supplementary Fig. 10). Moreover, regardless of the wide manipulation of pH from 4 to 10, the extraction efficiency of U(VI) on Ti(OH)PO$_4$ exceeded 90% (Fig. 3c). Furthermore, the

anti-interference capability of Ti(OH)PO$_4$ was demonstrated by extraction efficiency in the presence of co-existing ions, especially SO$_4^{2-}$, CO$_3^{2-}$, Cl$^-$, C$_2$O$_4^{2-}$, NO$_3^-$, and NH$_4^+$, which commonly existed in the wastewater of nuclear production (Fig. 3d). We also tested the changes in pH and ion concentrations before and after the electrochemical uranium extraction reaction (Supplementary Fig. 11). All the pH values of the electrolyte increased after the electrochemical reaction because of the hydrogen evolution in the electrochemical process. The extraction efficiency of U(VI) remained at a high level of >98%, indicating that none of the interference ions exhibited obvious influence on the U(VI) extraction. Moreover, the concentrations of anions in the electrolyte displayed a negligible decrease after the electrochemical reaction, which was attributed to the slight adsorption of ions by the electrode. We also evaluated the influence of F$^-$ concentration change on uranium recovery (Fig. 3e). Regardless of the initial concentration of F$^-$, the U(VI) concentration was reduced from approximately 100 mg L$^{-1}$ to ~1 mg L$^{-1}$ after the reaction with a high level of >98%, indicating that the F$^-$ concentrations did not exhibit an obvious influence on the U(VI) extraction for the Ti(OH)PO$_4$ electrode (Supplementary Fig. 12a). Such a phenomenon was attributed to the selective binding of UO$_2$F$_x$ by ion pair sites, instead of UO$_2^{2+}$. Furthermore, slight decreases in the concentration of F$^-$ were observed after the electrochemical reaction, which was ascribed to the adsorption by the Ti$^{\delta+}$ cation site and further co-crystallization by the uranium product (Supplementary Fig. 12b). Moreover, as the initial concentration of U(VI) varied from 50 mg L$^{-1}$ to 600 mg L$^{-1}$, the extraction efficiency of U(VI) by Ti(OH)PO$_4$ remained consistently above 99%, indicating an extraction capacity of 5950 mg g$^{-1}$ without the phenomenon of extraction saturation (Fig. 3f). These results suggested the potential use of Ti(OH)PO$_4$ with ion pair sites towards the electrochemical uranium extraction in the wastewater of nuclear production.

## Evolution of uranium species

The effective electrochemical uranium extraction of Ti(OH)PO$_4$ nanorods with ion pair sites in the presence of high concentrations of F$^-$ inspired us to explore the process of uranium deposition on the electrode. The time-dependent extraction experiment on Ti(OH)PO$_4$ nanorods was conducted in an aqueous electrolyte containing 600 mg L$^{-1}$ of U(VI) and 30 g L$^{-1}$ of F$^-$ (Fig. 4a). At the time point of 1 h, the deposited species appeared and exhibited a gray color. With the reaction proceeding to 3 h and 5 h, the gray deposit was gradually transformed into the yellow product, together with the aggregation of the solid deposits on the electrode. The color of the deposit finally transformed to a stable state of deep yellow at 7 h. To validate the species during the uranium extraction, we collected the solid deposits at different time points. As indicated by XRD patterns, the gray deposits at 1 h and 3 h were ascribed to U$_3$O$_7$ (JCPDS #42-1215), which was a special species of UO$_{2+x}$ containing U(V) and tetravalent uranium (U(IV)) (Fig. 4b). The existence of U(IV) can be explained by the dismutation of U(V). By comparison, the XRD of yellow deposits at 5 h and 7 h well fitted by K$_3$UO$_2$F$_5$ (JCPDS #38-0023) with domain species of U(VI), indicating the oxidation of low valent uranium in the deposits during the reaction. The K$_3$UO$_2$F$_5$ species was also responsible for the slight decreases in the concentration of F$^-$ in the experiment of Supplementary Fig. 12. The K in the final uranium extraction product (K$_3$UO$_2$F$_5$) originated from the KF in the electrolyte, which could be replaced by Na in Na$_3$UO$_2$F$_5$ when using NaF as electrolyte (Supplementary Fig. 13).

The result of uranium evolution was verified by the U 4$f$ XPS spectra at different time points (Fig. 4c). With the reaction proceeding to 3 h, the peaks of U 4$f_{7/2}$ were negatively shifted from 381.5 eV to 380.7 eV, indicating the continuous reduction of uranium during the reaction. With the reaction proceeding to 4 h and 7 h, the gray deposit was gradually transformed into the yellow product, together with the peaks of U 4$f_{7/2}$ positively shifted to 381.8 eV, indicating the oxidation

of low valent uranium in the deposits during the reaction[37,38]. For a precise interpretation, the gray deposits possessed more content of U(IV) and U(V) than that of the yellow deposits (Supplementary Fig. 14). The intermediate of U$_3$O$_7$ and final product of K$_3$UO$_2$F$_5$ were also observed through the deposits of multiple cycles of CV test, which verified the evolution process of uranium species (Supplementary Fig. 15). The alternation of the uranium species in the deposit was accompanied by the shape evolution. At a reaction time of 1 h, the initial gray uranium deposits displayed a nanoneedle morphology with low crystallization (Supplementary Fig. 16). As time proceeded, the nanoneedles gradually evolved into nanosheets, with visible crystalline lattice fringes. Figure 4d showed a typical TEM image of the final yellow deposits, which showed the flexible nanosheet morphology. Moreover, the lattice fringes of this nanosheet possessed the interplanar spacing of 0.32 nm, belonging to the (220) facet of K$_3$UO$_2$F$_5$ (Fig. 4e). In the EDS measurement, the U, K, and F elements were uniformly distributed in the nanosheets, further demonstrating the composition of the K$_3$UO$_2$F$_5$ in the final yellow deposits (Fig. 4f). Accompanied by the shape evolution of the uranium deposit, the morphology of Ti(OH)PO$_4$ transformed from pristine stacked nanosheets to exfoliated nanosheets through the expansion of interlayer spacing (Supplementary Fig. 17). This result demonstrated the uranyl intercalation into the interlayer spaces of Ti(OH)PO$_4$ during the electrochemical reaction, which verified the advantage of larger interlayer spacing of Ti(OH)PO$_4$ than the diameter of UO$_2$F$_x$.

To validate the species evolution involved in the uranium extraction, we directly investigated the valence state and coordination environment of uranium in the electrode at 3 h during the electrochemical reaction. Notably, the key intermediate deposit was not separated from the extraction material in this case. As shown by the U $L_3$-edge XANES spectra (Fig. 4g), the characteristic peaks at 17175.7 eV in the spectrum were identified as the white line peak, corresponding to the transition of electrons from the occupied U 2$p$ orbital to the unoccupied U 6$d$[39]. Significantly, the absorption edge position of electrochemical products was situated between UO$_2$(NO$_3$)$_2$ and U$_3$O$_8$, with a slight shift towards the low-E side. This observation implied that the valence states of U cations encompassed a range between +4 and +6, indicating the uranium extraction underwent the electrochemical reduction process. Figure 4h illustrated the Fourier transform (FT) $k^2$-weighted extended X-ray absorption fine structure (EXAFS) spectra of the electrochemical product. The FT curve of the electrochemical product had two main peaks at 1.4 and 1.9 Å, which were attributed to the U-O coordination of the first and second shells[40]. The first and second shells were denoted as U-O$_{ax}$ and U-O$_{eq}$, which respectively referred to the axial O of uranyl species and the O atom at the adsorption site of ion pair sites. Through the EXAFS fitting, the coordination numbers of U-O$_{ax}$ and U-O$_{eq}$ were 2.2 and 3.4, respectively (Fig. 4i and Table 1). This result suggested that the adsorption process was assisted by three O atoms in the surface ion pair sites of Ti(OH)PO$_4$ nanorods. Specially, the U-F bond co-existed with a coordination number of 1.1, demonstrating the target adsorbate involved in the uranium extraction was the overall UO$_2$F$_x$ instead of UO$_2^{2+}$, which resulted from the Coulomb interaction between ion pair sites and UO$_2$F$_x$.

## Reaction mechanism

The unusual adsorption of the overall UO$_2$F$_x$ and the corresponding species evolution motivate us to explore the intrinsic reaction mechanism. Based on the species evolution, we constructed a scheme to simulate the reaction process (Fig. 5a). The UO$_2$F$_x$ was adsorbed on the ion pair sites of Ti(OH)PO$_4$ nanorods without the separation of the U-F bond. After that, the uranyl was reduced to low-valent species, which weakened the Coulombian force of uranyl and F$^-$, thus resulting in the formation of an intermediate UO$_{2+x}$ product. The metastable low-valent uranium was then oxidized, followed by crystallizing with F$^-$ and forming the final K$_3$UO$_2$F$_5$.

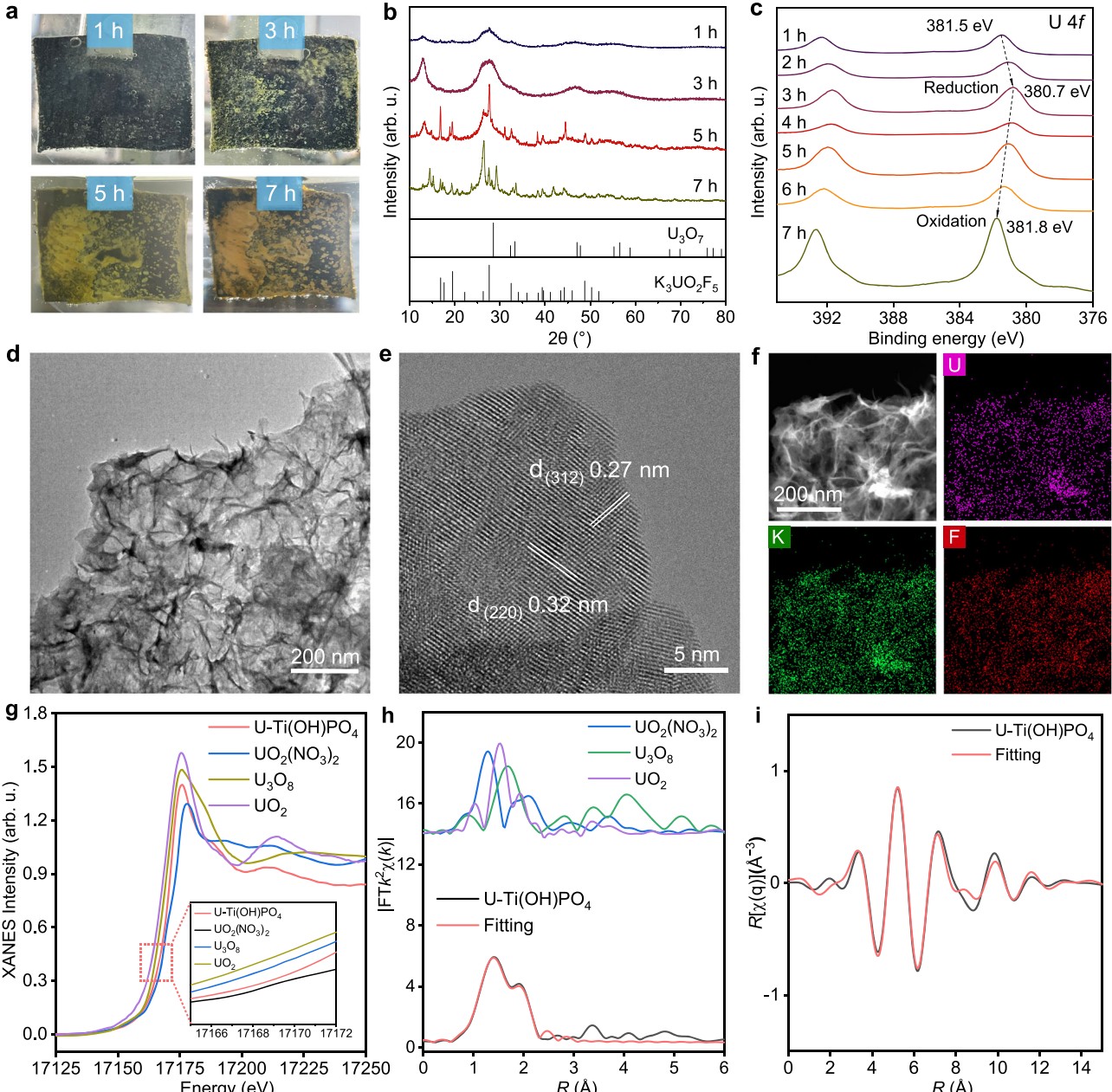

**Fig. 4 | Evolution of uranium species. a** Photographs of the evolution of uranium species during the uranium extraction. **b** XRD pattern of the collected solid deposits at different time points. **c** U 4*f* XPS spectrum of U-Ti(OH)PO₄ electrodes at different electrochemical extraction time points. **d** TEM image of the final yellow deposits with nanosheet morphology. **e** HRTEM and **f** EDS measurement of the composition of the K₃UO₂F₅ in the final yellow deposits. **g** The U *L₃*-edge XANES spectra of intermediate deposit. Inset: magnified pre-edge XANES region. **h** The FT *k²*-weighted EXAFS spectra of the electrochemical product. **i** The corresponding *k*-space fitting curves of the electrochemical product. Source data are provided as a Source Data file.

Considering the adsorption of the overall UO₂₊ₓ was the key factor that influenced the extraction efficiency of uranium, we theoretically simulated the adsorption of UO₂Fₓ on the ion pair site of Ti(OH)PO₄ nanorods (Fig. 5b, Supplementary Table 3, and Supplementary Data 1).

**Table. 1 | Structural parameters of U-Ti(OH)PO₄ at the U *L₃*-edge extracted from quantitative EXAFS curve-fittings**

| Path | CN | R (Å) | σ² (10⁻³ Å²) | ΔE (eV) | R-factor |
|------|-----|-------|--------------|---------|----------|
| U-O$_{ax}$ | 2.2 | 1.7 | 4.1 | −11.1 | 0.004 |
| U-O$_{eq}$ | 3.4 | 2.3 | 4.0 | 6.8 | 0.004 |
| U-F | 1.1 | 2.1 | 5.0 | −11.0 | 0.004 |

Taking UO₂F⁺ as an example, we compared the optimized adsorption configuration on OH-terminated Ti₃C₂ and Ti(OH)PO₄ with Ti^δ⁺ on the surface. The OH-terminated Ti₃C₂ possessed the normal −OH sites for bare UO₂²⁺ adsorption, which possessed adsorption energy of −2.6 eV for UO₂F⁺ without the interaction of F⁻ and Ti⁴⁺. For the Ti(OH)PO₄ model, the coordination of surface Ti^δ⁺ was unsaturated, giving rise to the interaction with F⁻. As such, the UO₂F⁺ has stabilized on the PO₄³⁻-Ti^δ⁺ ion pair sites through both the PO₄³⁻-UO₂²⁺ and Ti^δ⁺-F⁻ interactions. In this model, three O atoms in the Ti(OH)PO₄ participated in the adsorption of UO₂F⁺, consistent with the experimental EXAFS result. Moreover, the adsorption energy of UO₂F⁺ on the ion pair sites of Ti(OH) PO₄ was −4.5 eV, which was much more negative than that on the −OH sites of OH-terminated Ti₃C₂. The adsorption energy was able to be

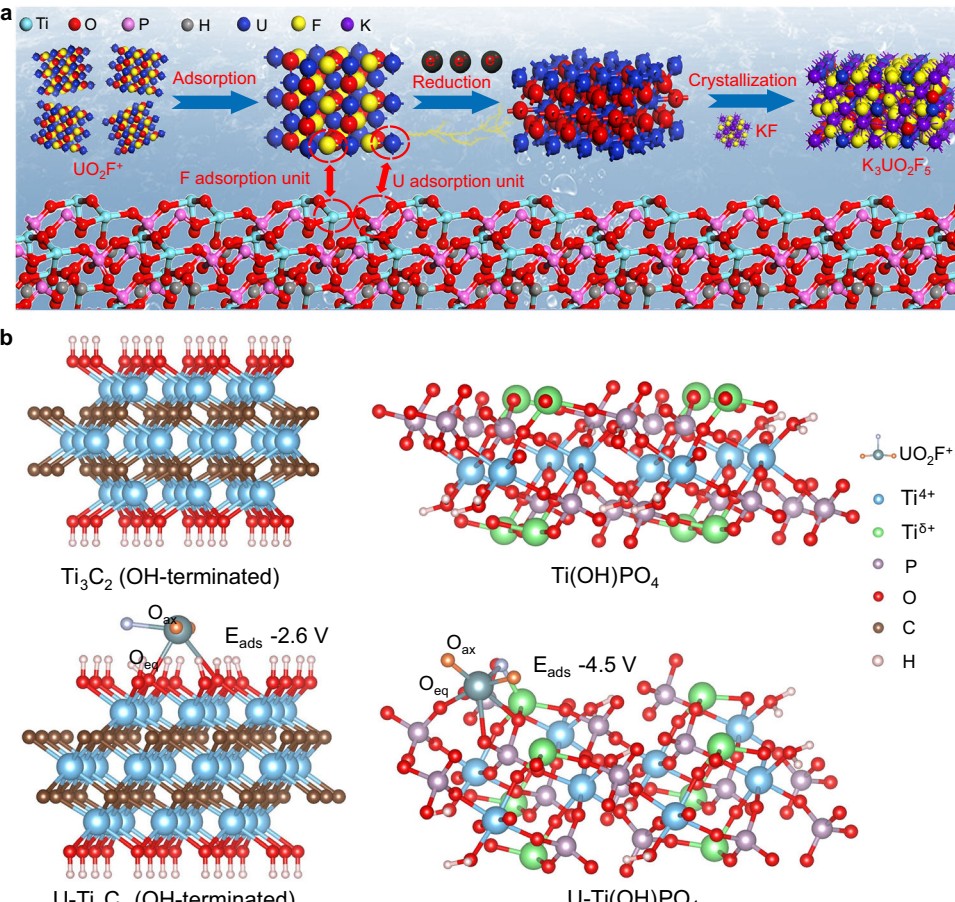

**Fig. 5 | Reaction mechanism of ion pair sites. a** The schematic diagram of the rational reaction mechanism of Ti(OH)PO₄ ion pair sites. **b** Optimized adsorption configurations of UO₂F⁺ adsorbed on the Ti₃C₂ (OH-terminated) and Ti(OH)PO₄. O$_{ax}$ the axial O of uranyl species, O$_{eq}$ the O atom at the adsorption site, E$_{ads}$ the adsorption energy of UO₂F⁺. Source data are provided as a Source Data file.

further decreased by introducing additional O vacancies (Supplementary Fig. 18). Furthermore, the bond angle of O$_{ax}$-U-O$_{ax}$ of UO₂F⁺-Ti(OH)PO₄ model was 100.6°, which was lower than that (167.4°) of UO₂F⁺-Ti₃C₂ (OH-terminated). Given that the reduction of UO₂²⁺ requires the decrease of bond angle, the ion pair sites of Ti(OH)PO₄ can facilitate the reduction of UO₂²⁺ in UO₂F⁺. As a result, the ion pair sites stabilized the adsorption of UO₂F$_x$ and promoted the reduction of UO₂²⁺, which was responsible for the uranium extraction in high concentrations of F⁻.

## Experiment of real nuclear wastewater

Motivated by the distinctive effect of ion pair sites on the uranium extraction in high concentrations of F⁻, we explored the uranium extraction from real nuclear wastewater on Ti(OH)PO₄. This wastewater consisted of high levels of F⁻ (8 g L⁻¹), CO₃²⁻ (5.5 g L⁻¹), HCO₃⁻ (4.5 g L⁻¹), NO₃⁻ (3 g L⁻¹), Cl⁻ (2 g L⁻¹), NH₄⁺ (3 g L⁻¹), and C₂O₄²⁻ (5 g L⁻¹), which was produced by the washing of container in a nuclear fuel production facility. After 7-h electrolysis at a constant current density of 30 mA cm⁻², the concentration of U(VI) was reduced from an initial value of 685.1 mg L⁻¹ to 2.2 mg L⁻¹, with 99.6% of extraction efficiency and 6829 mg g⁻¹ of extraction capacity (Fig. 6a). Such value of extraction efficiency was 1.7-time larger relative to that of OH-terminated Ti₃C₂ (55.9%), indicating the effectiveness of Ti(OH)PO₄ with ion pair sites (Supplementary Fig. 19). With the proceeding of the electrolysis, the yellow color of real wastewater gradually faded, finally resulting in a colorless and transparent residual liquid (Fig. 6b). The concentration of all metal species in the transparent residual liquid was displayed in the Supplementary Table 4. Compared with the yellow real

wastewater, the K⁺ and Na⁺ respectively decreased by 697.3 mg L⁻¹ (0.5% decrease) and 136.6 mg L⁻¹ (2.2% decrease), which was mainly ascribed to the adsorption of the electrode, instead of the electrochemical extraction (Supplementary Fig. 20). The majority of adsorbed Na⁺ and K⁺ could be removed from the electrode by washing process. Additionally, the low initial concentrations (below 2 mg L⁻¹) of trace metal elements were essentially inconsequential to the uranium extraction process and purity of uranium deposits. Therefore, the uranium was able to be efficiently extracted from the electrode.

After the electrochemical reaction, the uranium was densely deposited on the electrode, which can be desorbed, simply precipitated by KOH, and collected as a solid powder (Fig. 6c). We analyzed the proportion of metal species in this powder by inductively coupled plasma optical emission spectrometry (ICP-OES) measurement (Fig. 6d). The proportion of uranium reached 92.1% among the metal elements in the powder, which was convenient for the following separation and reuse of uranium. In addition, we employed a typical precipitation-calcination process for the further purification of uranium products (Fig. 6e). Initially, we collected and dissolved uranium extraction product in HCl. Subsequently, excess ammonia was added to the above solution to form a precipitate of ammonium diuranate. Finally, ammonium diuranate was collected by centrifugation and heated at 550 °C for 5 h to form a black-green uranium product. According to XRD (Supplementary Fig. 21) and ICP-OES tests (Fig. 6f), the purified black-green uranium product was identified as U₃O₈ (JCPDS #31-1425) with a uranium content of 99.93% among the metal elements, which can be directly used in the uranium production.

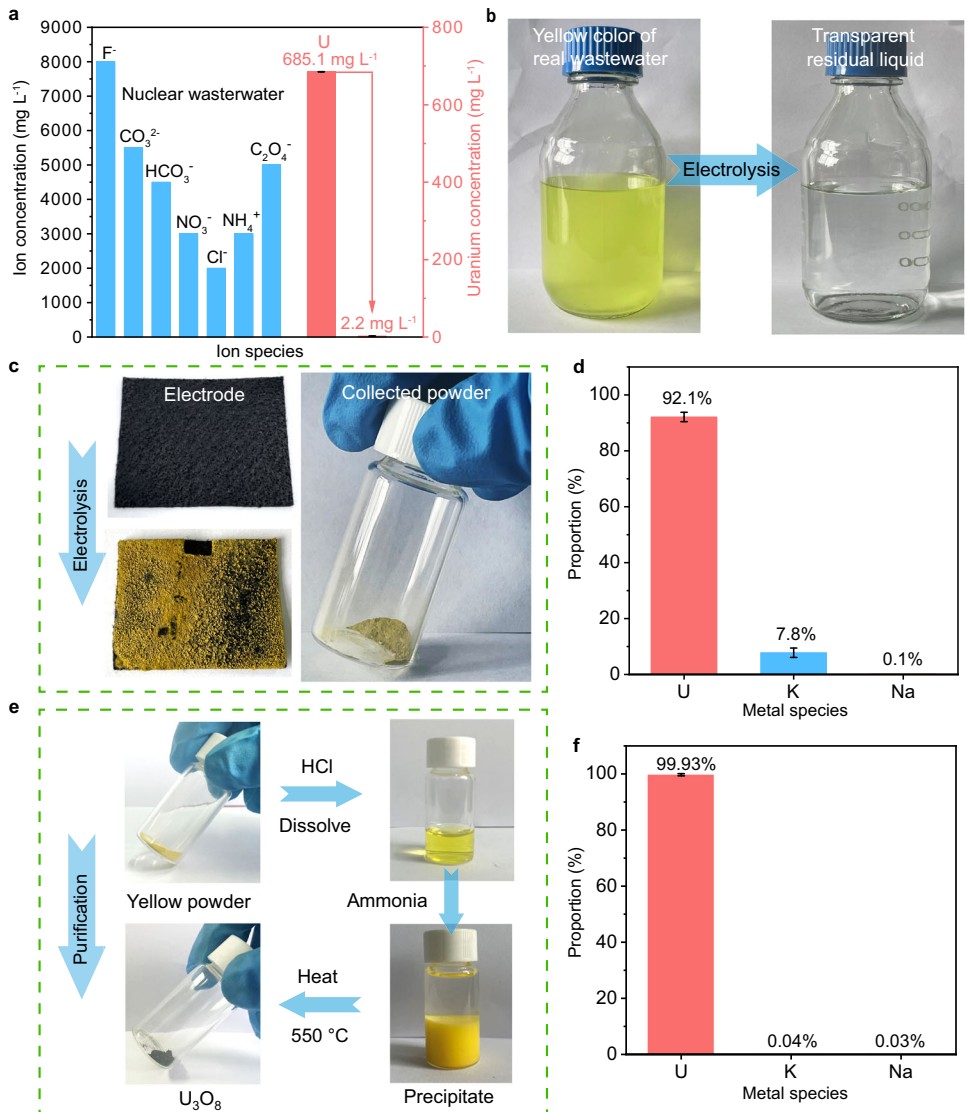

**Fig. 6 | Experiment of real nuclear wastewater of ion pair sites. a** The electrochemical extraction efficiency of U(VI) on Ti(OH)PO₄ in real nuclear wastewater. **b** The color change of real nuclear wastewater before and after electrolysis. **c** The deposited uranium on the electrode and the corresponding collected powder after electrolysis. **d** The proportion of uranium among the metal species in the powder. **e** The schematic diagram of the precipitation-calcination purification process for uranium. **f** The proportion of uranium among the metal species in the purification product. Error bars represent standard deviation of three measurements. Source data are provided as a Source Data file.

Consequently, the electrochemical uranium extraction by Ti(OH)PO₄ provided a feasible strategy for resource recovery in real wastewater of nuclear production. We also evaluated the approximate operating cost. The consumed electricity was 31.5 Wh, which means the extraction per kg of U requires electricity of ~115 kWh, corresponding to an electricity cost of ~8 US dollars (according to the price of China). For the cost of Ti(OH)PO₄ electrode, the price was approximately 0.2 US dollars considering the purchase of Ti₃AlC₂ precursor, synthetic material, and graphite support. Considering the recycling use of the electrode, the cost of electrode material was ~14 US dollars for extracting 1 kg of uranium. Therefore, the overall cost was approximately 22 US dollars for the extraction of 1 kg U considering both energy consumption and material costs.

## Discussion

In summary, the Ti$^{\delta+}$-PO₄$^{3-}$ ion pair sites are constructed in Ti(OH)PO₄ as uranium extraction material for nuclear wastewater containing high concentrations of F⁻ from fuel production. In simulated electrolytes with up to 30 g L⁻¹ of F⁻, Ti(OH)PO₄ possesses an extraction efficiency of 95.1% in 5 h. The mechanistic study reveals that UO₂Fₓ can be strongly bound in the form of 2O$_{ax}$-U-3O$_{eq}$ and Ti-F bonds. With the reaction proceeding, the adsorbed UO₂Fₓ is transformed from the reduced state (U₃O₇) to the crystalline state (K₃UO₂F₅). In a real nuclear wastewater experiment, the Ti(OH)PO₄ exhibits 99.6% extraction efficiency of uranium within 7 h regardless of the interferential 8 g L⁻¹ of F⁻, with an extraction capacity of 6829 mg g⁻¹ without saturation. After the simple collection, the powder product containing high-purity of uranium is obtained, demonstrating the successful recycling of uranium in real wastewater. Our work not only presents an efficient uranium extraction material for resisting high concentrations of F⁻, but also provides an efficient strategy for uranium recovery in real and complex nuclear wastewater.

## Methods

### Chemicals and materials

Ti₃AlC₂ powder (400 mesh) was purchased from 11 Technology Co., Ltd. Hydrochloric acid (HCl, ≥ 36% purity) and Phytic acid (PA, 70%

purity) were supplied by Chengdu Kelong Chemical Factory. Lithium fluoride (LiF, 99% purity) and Uranium nitrate ($UO_2(NO_3)_2 \cdot 6H_2O$, 99% purity) were obtained from Aladdin Shanghai Chemical. Graphite felt (Carbon content ≥ 99 %) and ethanol (AR) were purchased from CeTech Co., Ltd. Nafion membrane solution (NR50, 5 wt%) was acquired from Shanghai Macklin Biochemical Co., Ltd. The real nuclear wastewater was collected from a nuclear fuel production facility. All reagents and solvents were used without purification.

## Characterizations

Scanning electron microscopy (SEM) images were displayed on a JSM-7800F Scanning Electron Microscope. Transmission electron microscopy (TEM), high-resolution transition electron microscopy (HRTEM), and energy dispersive X-ray spectroscopy (EDS) images were displayed on a JEM 2100 F transmission electron microscope operating at an accelerating voltage of 200 kV. Atomic force microscopy (AFM) images were obtained on a Bruker Dimension ICON testing at ScanAsyt mode. Inductively coupled plasma optical emission spectrometry (ICP-OES) analyses were performed on PerkinElmer ICP 2100 and PerkinElmer Optima 5300 DV. X-ray diffraction (XRD) data were collected on a Smartlab SE diffractometer equipped with a Cu Kα source. Raman spectra (Raman) were obtained on a Thermo Fischer DXR Laser Raman spectrometer testing at a wavelength of 532 nm. Fourier transform infrared (FT-IR) spectroscopy was determined from Nicolet iS 10 Spectrometer. The defects and surface properties of materials were identified by electron spin resonance (ESR) spectroscopy using a Bruker EMXplus. X-ray photoelectron spectroscopy (XPS) analysis was performed through Thermo Kalpha, equipped with a monochromatic Al Kα X−ray source. The concentration of anions in the electrolyte was determined by an ion chromatograph (IC). The oxidation states characterization of the samples was utilized by the X-ray absorption fine structure (XAFS) through the transmission mode for Ti $L$ edge and U $L_3$-edge in the National Synchrotron Radiation Laboratory (NSNR). The absorption edge positions were calibrated during the XAFS measurements. Extended X-ray absorption fine structure (EXAFS) data were collected from the ATHENA module, integrated within the IFEFIT software suite. Linear sweep voltammetry (LSV) and Cyclic Voltammetry (CV) data were displayed on the CHI 660E electrochemistry workstation.

## Theoretical calculations

We have performed the DFT calculations in our system by using the Vienna ab initio simulation package (VASP, Version: VASP 5.4.4)[41,42]. The PBE function was used for describing electronic exchange[43]. The projector augmented wave method was adopted to describe the interactions between the ion cores and valence electrons[44,45]. The energy cutoff for the plane-wave basis set was 500 eV. The Brillouin scheme sampled the Monkhorst-Pack scheme using a $3 \times 3 \times 1$ k-point grid[46]. During the geometry optimization and electronic structure calculation, the atomic positions were optimized until the energy and the maximum force were $<10^{-5}$ eV atom$^{-1}$ and 0.02 eV Å$^{-1}$, respectively. The van der Waals dispersion by employing the DFT-D3 method of Grimme was considered for all the calculations[47].

## Construction of Ti(OH)PO₄ with ion pair sites

The exfoliated lamellar OH-terminated $Ti_3C_2$ nanosheets were synthesized following the MILD method by in situ HF selective etching of the Al layer of $Ti_3AlC_2$ precursor. Specifically, LiF (3.20 g) was dissolved with HCl (12.0 M, 40 mL) solution in a Teflon beaker, followed by the slow addition of $Ti_3AlC_2$ (2.00 g) in an ice bath under magnetic stirring. The reaction was conducted at 40 °C under ventilated conditions for 24 h to avoid safety hazards caused by HF use. The accordion-like stack -F-terminated $Ti_3C_2$ was collected and centrifuged at 2515 × g several times with deionized water to remove any vestigial acid until the supernatant reached a pH of 6. Subsequently, the slurry above was

immersed in deionized water (100 mL) to form a homogeneous suspension via sonication for 2 h under the protection of $N_2$. After that, the exfoliated OH-terminated $Ti_3C_2T_x$ aqueous supernatant in a centrifuge cube was collected after centrifugation once again at 1760×$g$ for 5 mins. Finally, the suspension for the experiments was prepared by diluting the exfoliated OH-terminated $Ti_3C_2T_x$ suspension to 20 mg mL$^{-1}$. The Ti(OH)PO₄ ion pair sites were synthesized via a facile wet chemical method. Typically, the exfoliated OH-terminated $Ti_3C_2T_x$ suspension (2 mL) was mixed with deionized water (40 mL) and stirred for 30 mins at room temperature to obtain a uniformly dispersed solution. Then, phytic acid (1.6 mL) was added to the above-dispersed solution dropwise and kept stirring for another 1 h. Afterward, the homogeneous solution was placed into a Teflon-lined autoclave (50 mL) and heated at 180 °C for 12 h. Finally, the Ti(OH)PO₄ ion pair sites were collected after washing with deionized water and ethanol several times, and freeze-dried for 24 h.

## Electrochemical uranium extraction experiments in simulated nuclear wastewater

The electrochemical uranium extraction experiments in simulating nuclear wastewater were measured with a three-electrode system. The working electrodes were fabricated as follows: Ti(OH)PO₄ ion pair sites (5.00 mg), carbon black (2.00 mg), and Nafion Membrane solution (20 μL) were mixed with ethanol (2 mL) and sonicated until they became a homogeneous ink. The as-prepared ink was directly brush-coated on graphite felt (1 cm × 1 cm) and dried at 80 °C. The platinum (Pt) wire and Ag/AgCl electrode were used as the counter electrode and reference electrode, respectively. The uranium extraction experiments were conducted under a constant current of 30 mA cm$^{-2}$. The electrochemical uranium extraction performances were tested in an aqueous solution (50 mL) containing 100 mg L$^{-1}$ of U(VI) and 30 g L$^{-1}$ of F$^-$ (F$^-$ were sourced from KF electrolyte). The non-voltage adsorption experiments were tested on an unpowered three-electrode system in 50 mL of simulating nuclear water. The as-prepared Ti(OH)PO₄ electrode, Pt wire, and Ag/AgCl electrode were also used as the working electrode, counter electrode, and reference electrode respectively. The condition of the adsorption test was similar to that of electrochemical extraction except for excluding external current or voltage. The effects of the initial concentration of uranium (from 50 to 600 mg L$^{-1}$), pH of the solution (from 4 to 10), interfering ions (including $SO_4^{2-}$, $CO_3^{2-}$, $Cl^-$, $C_2O_4^{2-}$, $NO^{3-}$, $NH_4^+$), and the influence of F$^-$ (from 5 to 30 g L$^{-1}$) on the electrochemical uranium extraction performance were measured in detail. The concentrations of the uranium and other cations in the electrolyte before and after electrolysis were determined by ICP-OES. The concentrations of the anions in the electrolyte before and after electrolysis were determined by IC. In the desorption process, the working electrode with extracted uranium was transferred into HCl aqueous solution (0.1 mol). The extraction experiments were conducted three times and the error bars were used in the curves. LSV and CV tests were performed in simulated nuclear wastewater at a scan rate of 10 mV s$^{-1}$.

## Electrochemical uranium extraction experiments in real nuclear wastewater

The electrochemical uranium extraction experiments in real nuclear wastewater were measured with a typical two-electrode system. Graphite felt was cut into 5 cm × 5 cm shapes as electrode substrates. Ti(OH)PO₄ ion pair sites (40.0 mg), carbon black (8.0 mg), and Nafion Membrane solution (80 μL) were mixed with ethanol (8 mL) and sonicated for 1 h to form a homogeneous ink. The as-prepared ink was then uniformly brush-coated on graphite felt to prepare the working electrode, and Pt wire was used as the counter electrode. The uranium extraction was operated under a constant current of 30 mA cm$^{-2}$ in 400 mL of real nuclear wastewater. The deposited electrode was immersed in ethanol and purified through an ultrasonic reaction, and

the powder was collected by drying. The proportions of metallic species in this powder were measured and analyzed by ICP-OES. The concentration changes of the uranium and other cations in real nuclear wastewater before and after electrolysis were determined by ICP-OES.

## Uranium product purification

The purification of uranium was performed through a typical precipitation-calcination process. Initially, the uranium product from the electrochemical extraction was collected and dissolved in HCl. Subsequently, excess ammonia was added to the above solution to form a precipitate of ammonium diuranate. Finally, ammonium diuranate was collected by centrifugation and heated at 550 °C for 5 h to form a black-green uranium product.

## Calculations

For quantifying the extraction efficiency ($n$, %), the uranium extraction capacity ($q_e$, mg g$^{-1}$), and adsorption energy ($E_{ads}$, eV) were calculated according to the following equations:

$$n = \frac{(C_0 - C_e)}{C_0} \quad (1)$$

Where $C_0$ (mg L$^{-1}$) and $C_e$ (mg L$^{-1}$) represent the initial and final uranium concentration, respectively.

$$q_e = \frac{(C_0 - C_e) \times V}{m} \quad (2)$$

Where $C_0$ (mg L$^{-1}$) and $C_e$ (mg L$^{-1}$) represent the initial and final uranium concentration, respectively; $m$ (g) and $V$ (L) denote the mass of the catalyst and volume of the electrolyte, respectively.

$$E_{ads} = E_{AS} - (E_S + E_A) \quad (3)$$

Where $E_{AS}$, $E_S$, and $E_A$ are the total energies of the adsorbate-substrate (AS), substrate (S), and adsorbate (A), respectively.

## Data availability

The data that supports the findings of the study are included in the main text and supplementary information files. Source data are provided with this paper. Source Data files are available in Figshare under accession code https://doi.org/10.6084/m9.figshare.25603905. Source data are provided with this paper.

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

## Acknowledgements

This work was supported by NSFC (No. U23A20105, R.H.; 22303002, Y.L.; 22106126, T.C.; and U2267224, W.Z.), the Open Fund of CNNC Key Laboratory for Uranium Extraction from Seawater (KLUES202201, W.Z.), the Project of State Key Laboratory of Environment-Friendly Energy Materials in SWUST (No. 20fksy19, W.Z.). Numerical computations were performed at Hefei Advanced Computing Center. The authors extend their gratitude to Shiyanjia Lab (www.shiyanjia.com) for providing invaluable assistance with the TEM and XPS analysis.

## Author contributions

T.L. made the most important contributions, although all authors made contributions to the work. T.L., R.H., and W.Z. designed the studies and wrote the paper. T.L. and H.J. performed most of the experiments. T.C., H.Z., and K.H. provided advice and reagents. C.J. and Y.L. performed DFT calculations. R.H. and W.Z. supervised the research. All authors contributed to data analysis and commented on the manuscript.

## Competing interests

The authors declare no competing interests.
