## [Peer Review File · Nature Communications]

Ion pair sites for efficient electrochemical extraction of uranium in real nuclear wastewaterREVIEWER COMMENTS

Reviewer #1 (Remarks to the Author):

The paper reports on the effective electrochemical uranium extraction of $\text{Ti}(\text{OH})\text{PO}_4$ nanorods with $\text{Ti}\delta^+-\text{PO}_4^{3-}$ ion pair extraction sites from nuclear wastewater containing high concentrations of F^- . Using the feasible strategy of ion pair extraction sites, the authors electrochemically extracted 99.6% of uranium from real nuclear wastewater within 7 h. The manuscript is well organized and written. However, there are some comments that authors should take into account before publication:

1. More discussion is needed on the etching and exfoliation protocol to prepare lamellar OH-terminated Ti_3C_2 nanosheets. The authors should clearly mention the product obtained (whether Ti_3C_2 stack or Ti_3C_2 lamellar nano sheets) after the selective removal of Al layers from Ti_3AlC_2 MAX using LiF/HCl .
2. Why did the authors negate or discuss the influence of “-F” termination group over the surface of $\text{Ti}_3\text{C}_2\text{Tx}$ while discussing the $\text{Ti}(\text{OH})\text{PO}_4$ with $\text{Ti}\delta^+-\text{PO}_4^{3-}$ ion pair extraction sites?
3. What is the significance of the larger interlayer spacing (0.725 nm) of $\text{Ti}(\text{OH})\text{PO}_4$ nanorods towards the electrochemical extraction of U(VI)?
4. During electrochemical deposition of U(IV), did the authors observe the intermediate formation through CV analysis? If so discuss.
5. The authors should also provide the efficacy of as-synthesized OH-terminated Ti_3C_2 nanosheets towards the electrochemical extraction of U(VI) for comparison.
6. What was the proportion of metal species of colorless and transparent residual liquid after U(VI) adsorption?
7. Authors should provide detailed XPS interpretation for Ti 2p of OH-terminated Ti_3C_2 nanosheets with respect to that of $\text{Ti}(\text{OH})\text{PO}_4$. It is worth mentioning the oxidation states of Ti atoms (Ti, Ti^{2+} , Ti^{3+}).
8. The authors should provide the experimental conditions of U adsorption through non-voltage adsorption method.
9. What would be the approximate operating cost of your proposed method to extract per kg of Uranium?

Reviewer #2 (Remarks to the Author):

This manuscript reported on the application of electrochemical uranium extraction from nuclear wastewater recycling uranium resources. It constructs $\text{Ti}\delta^+-\text{PO}_4^{3-}$ ion pair extraction sites in $\text{Ti}(\text{OH})\text{PO}_4$ for efficient electrochemical uranium extraction in wastewater from nuclear fuel production. These sites can selectively bind with UO_2Fx through the combined Ti-F bond and multiple O-U-O bonds. In real nuclear wastewater, the electrochemical extraction efficiency of uranium reaches 99.6% within 7 h, corresponding to a remarkable extraction capacity of 6829 mg g^{-1} without saturation. However, some aspects of the current manuscript should be improved. Thus, I recommend major revision before its

publication.

1. In Fig. 2, the Ti3C2 needs to be characterized by Fourier transform infrared to prove the change of its surface functional groups.
2. When studying the influence of other ions and pH, it is necessary to detect the changes in ion concentration and pH before and after the reaction.
3. The author emphasized the influence of F⁻ on uranium recovery in the front, but did not explain in detail the change and role of F⁻ in uranium recovery in this study. The author should supplement the influence of F⁻ concentration change on uranium recovery and the change of F⁻ ion concentration before and after recovery.
4. The author should provide more direct proof for the change of uranium valence state.
5. The composition of electrolyte should be indicated in the experimental method, whether the K element in K₃UO₂F₅ comes from electrolyte, whether the change of electrolyte will change the product, and whether the K and F in the product can be further removed.
6. Please provide a more intuitive energy diagram of the splitting orbits of Ti 3d peak.

Point-by-point response to reviewer comments

Manuscript ID: NCOMMS-23-60509A

MS Type: Article

Title: "Ion pair sites for efficient electrochemical extraction of uranium in real nuclear wastewater"

First of all, we sincerely thank the editor and all Reviewers for giving us valuable and thoughtful comments to improve the quality of this manuscript. According to the Reviewers' suggestions, we have performed several additional experiments to improve the strength of the study and make the following points clear. We provide point-by-point replies to the Reviewers' comments below.

Reviewer #1

The paper reports on the effective electrochemical uranium extraction of $\text{Ti}(\text{OH})\text{PO}_4$ nanorods with $\text{Ti}^{\delta+}\text{-PO}_4^{3-}$ ion pair extraction sites from nuclear wastewater containing high concentrations of F^- . Using the feasible strategy of ion pair extraction sites, the authors electrochemically extracted 99.6% of uranium from real nuclear wastewater within 7 h. The manuscript is well organized and written. However, there are some comments that authors should take into account before publication:

Reply: We greatly appreciate your professional comments on our article. As you are concerned, several issues need to be resolved. Based on your suggestion, we have made extensive corrections to the previous manuscript, and the specific corrections are as follows.

1. More discussion is needed on the etching and exfoliation protocol to prepare lamellar OH-terminated Ti_3C_2 nanosheets. The authors should clearly mention the product obtained (whether Ti_3C_2 stack or Ti_3C_2 lamellar nanosheets) after the selective removal of Al layers from Ti_3AlC_2 MAX using LiF/HCl .

Reply: We genuinely thank the reviewer for raising this issue, which helps us to improve the quality of our manuscript. According to the reviewer's suggestion, we have supplemented and clarified the synthesis schematic diagram of the etching and exfoliation process of OH-terminated Ti_3C_2 nanosheets (Fig. R1). Initially, the Ti_3AlC_2 with the Al layers was purchased

and exhibited a bulk property. Subsequently, the Al layers were etched by HF (originated from HCl and LiF), followed by a combination of vigorous manual shaking and centrifugation. At this stage, the Ti_3C_2 displayed an accordion-like morphology, which was formed by the stack of nanosheets. Finally, after one hour of sonication under N_2 protection in the aqueous solution, the stack of Ti_3C_2 nanosheets was exfoliated as lamellar nanosheets, together with the substitution of surface -F group by -OH group. The OH-terminated Ti_3C_2 lamellar nanosheets were the precursor for the synthesis of $\text{Ti}(\text{OH})\text{PO}_4$.

We have incorporated the corresponding descriptive information within the main text section (Page 4, lines 79-82) and supplementary information.

Fig. R1 The synthesis schematic diagram of OH-terminated Ti_3C_2 nanosheets

Revision: Fig. R1 is added as Supplementary Fig. 1 in the revised Supplementary Information.

2. Why did the authors negate or discuss the influence of “-F” termination group over the surface of $\text{Ti}_3\text{C}_2\text{T}_x$ while discussing the $\text{Ti}(\text{OH})\text{PO}_4$ with $\text{Ti}^{\delta+}\text{-PO}_4^{3-}$ ion pair extraction sites?

Reply: We genuinely thank this reviewer for raising this issue. According to the Hard and Soft Acids and Bases (HSAB) theory, as a hard acid, uranyl exhibits a stronger binding affinity with OH^- than that with F^- . As such, in the comparison of $\text{Ti}(\text{OH})\text{PO}_4$, the OH-terminated Ti_3C_2 was especially noted, instead of F-terminated Ti_3C_2 . In addition, the O^{2-} in PO_4^{3-} of $\text{Ti}(\text{OH})\text{PO}_4$ possesses a harder basicity compared to -OH groups, which motivated the discussion of $\text{Ti}^{\delta+}\text{-PO}_4^{3-}$ ion pair extraction sites.

To further exclude the concerns of “-F” termination group as extraction sites, we have evaluated the electrochemical performance of Ti_3C_2 nanosheets with different amounts of residual “-F” termination group. The amount of residual “-F” termination group in OH-terminated Ti_3C_2 nanosheets was controlled by manipulating the pH value involved in the exfoliation process of accordion-like Ti_3C_2 stack. With the decrease of pH value, the amount of

residual “F⁻” termination group in OH-terminated Ti₃C₂ nanosheets gradually increased. Accordingly, we prepared three kinds of OH-terminated Ti₃C₂ nanosheets, namely poor F⁻, moderate F⁻, and rich F⁻. Determined by ion chromatography (IC), the mass content of F element was respectively 0.18 μg, 0.72 μg, and 0.92 μg for poor F⁻, moderate F⁻, and rich F⁻ in 5 mg of OH-terminated Ti₃C₂ nanosheets (Fig. R2a). Meanwhile, using X-ray photoelectron spectroscopy (XPS) tests which indicated the surface signal of elements (Fig. R2b and Table R1), the atomic ratio of F⁻ were 8.4%, 13.7%, and 20.1% for poor F⁻, moderate F⁻, and rich F⁻, respectively. For the performance of electrochemical uranium extraction (Fig. R2c), the extraction efficiency of U(VI) on OH-terminated Ti₃C₂ nanosheets with poor F⁻ was 57.5%, which outperformed that with moderate F⁻ (51.5%) and rich F⁻ (47.1%). This result indicated that the increase of -F termination decreased the uranium extraction efficiency of Ti₃C₂, demonstrating that the extraction sites in OH-terminated Ti₃C₂ nanosheets were -OH, instead of residual “F⁻”. Therefore, we considered OH-terminated Ti₃C₂ as the primary influential group.

We have added the corresponding discussion in the revised manuscript (Page 9, lines 186-188).

Fig. R2 (a) The mass content of F in the OH-terminated Ti₃C₂. (b) XPS spectrum of the OH-terminated Ti₃C₂ with different F contents. (c) The extraction efficiency of uranium in the OH-terminated Ti₃C₂ with different F contents.

Revision: Fig. R2 is added as Supplementary Fig. 7 in the revised Supplementary Information.

Table R1. XPS semi-quantitative data of F contents in OH-terminated Ti₃C₂ and the corresponding electrochemical uranium extraction performance.

OH-terminated Ti ₃ C ₂	F atomic (%)	Extraction efficiency (%)
Poor F ⁻	8.4	57.5
Moderate F ⁻	13.7	51.5
Rich F ⁻	20.1	47.1

Revision: Table R1 is added as Supplementary Table 2 in the revised Supplementary Information.

3. What is the significance of the larger interlayer spacing (0.725 nm) of Ti(OH)PO₄ nanorods towards the electrochemical extraction of U(VI)?

Reply: We sincerely thank this reviewer for raising this issue. The larger interlayer spacing (0.725 nm) of Ti(OH)PO₄ nanorods enabled the uranyl intercalation into the interlayer and thus benefited the exposure of extraction sites. To verify this point, we have explored the morphology transformation of Ti(OH)PO₄ during the electrochemical uranium extraction process and proposed the corresponding schematic diagram (Fig. R3). Initially, Ti(OH)PO₄ possessed an interlayer distance of 0.725 nm, which was larger than the dimensions of uranyl fluoride ions (below 0.4 nm). After 1 h of electrochemical uranium extraction, the interlayer distance of Ti(OH)PO₄ increased to 0.811 nm, which resulted from the uranyl intercalation into the interlayer spaces of Ti(OH)PO₄. With the reaction proceeding to 7 h, Ti(OH)PO₄ underwent further intercalation and exfoliation and transformed into a lamellar structure. Therefore, the large interlayer distance of Ti(OH)PO₄ facilitated the interlayered ion pair sites for uranyl fluoride adsorption and reduction.

We have added the corresponding discussion in the revised manuscript (Page 12, lines 270-275).

Fig. R3 The schematic diagram of uranyl intercalation and the subsequent exfoliation of Ti(OH)PO₄ during the electrochemical uranium extraction process.

Revision: Fig. R3 is added as Supplementary Fig. 17 in the revised Supplementary Information.

4. During electrochemical deposition of U(IV), did the authors observe the intermediate formation through CV analysis? If so discuss.

Reply: We thank the reviewer for this insightful suggestion. As suggested, we additionally performed Cyclic Voltammetry (CV) tests ranging from -1.2 to 0 V in the presence of U(VI) in 30 g L^{-1} of F^- with the scan rate of 100 mV s^{-1} . To validate the species during the uranium extraction, we collected the solid deposits at different CV cycles. As indicated by XRD patterns (Fig. R4a), the grey deposits at 3000 cycles were ascribed to U_3O_7 (JCPDS #15-0004), which was a special species of UO_{2+x} containing U(V) and U(IV). The existence of U(IV) can be explained by the disproportionation of U(V). By comparison, the XRD of yellow deposits at 10000 cycles well fitted by $\text{K}_3\text{UO}_2\text{F}_5$ (JCPDS #38-0023) with domain species of U(VI), indicating the oxidation of low valent uranium in the deposits during the reaction (Fig. R4b). The intermediate formation and the final product of CV tests were consistent with that of the galvanostatic process.

We have added the discussion on the intermediate of CV test in the revised manuscript (Page 12, lines 260-262).

Fig. R4 XRD pattern of the collected solid deposits at different CV cycles.

Revision: Fig. R4 is added as Supplementary Fig. 15 in the revised Supplementary Information.

5. The authors should also provide the efficacy of as-synthesized OH-terminated Ti_3C_2 nanosheets towards the electrochemical extraction of U(VI) for comparison.

Reply: We greatly appreciate your professional comments on our article. As the reviewer pointed out, we explored the uranium extraction from real nuclear wastewater on OH-terminated Ti_3C_2 for comparison (Fig. R5). After 7-h electrolysis at a constant current density of 30 mA cm^{-2} , the concentration of U(VI) was reduced from an initial value of 685.1 mg L^{-1} to 301.6 mg L^{-1} with an extraction efficiency of 55.9%. By comparison, the extraction efficiency of $\text{Ti}(\text{OH})\text{PO}_4$

(99.6%) in real nuclear wastewater was 1.7-time larger than that of OH-terminated Ti_3C_2 . Consequently, the electrochemical uranium extraction by $\text{Ti}(\text{OH})\text{PO}_4$ with ion pair sites provided a feasible strategy for uranium extraction in real wastewater of nuclear production.

We have added the discussion on the efficacy of as-synthesized OH-terminated Ti_3C_2 nanosheets towards the electrochemical extraction of U(VI) from real nuclear wastewater in the revised manuscript (Page 15, lines 340-342).

Fig. R5 The electrochemical extraction efficiency of U(VI) on OH-terminated Ti_3C_2 in real nuclear wastewater.

Revision: Fig. R5 is added as Supplementary Fig. 19 in the revised Supplementary Information.

6. What was the proportion of metal species of colorless and transparent residual liquid after U(VI) adsorption?

Reply: We appreciate the reviewer for providing these constructive comments to improve the quality of the article. According to the reviewer's suggestion, we measured the concentration of metal species in the yellow color of real wastewater and transparent residual liquid by ICP-OES and ICP-MS. As shown in Fig. R6a, the concentration of K decreased from an initial value of 141637.1 mg L⁻¹ to a final value of 140,939.8 mg L⁻¹ (decreased by 697.3 mg L⁻¹, 0.5%) after the reaction, whereas for Na, the concentration was reduced from an initial value of 6145.2 mg L⁻¹ to a final value of 6,008.6 mg L⁻¹ (decreased by 136.6 mg L⁻¹, 2.2%). Moreover, the concentrations of various other trace metal elements also exhibited a decrease after the electrochemical reaction (Fig. R6b). However, their low initial concentrations (below 2 mg L⁻¹) were essentially inconsequential to the uranium extraction process and purity of uranium deposits. Notably, the concentration of U decreased from an initial value of 685.1 mg L⁻¹ to 2.2

mg L⁻¹, with 99.6% extraction efficiency. Furthermore, we analyzed the transparent residual liquid after uranium extraction (Fig. R6c and Table R2), and the proportion of K, Na, U, Mn, Al, Fe, and Ca were 95.908828%, 4.0888222%, 0.0014971%, 0.0000953%, 0.0000157%, 0.0006805%, and 0.0000612%, respectively. Notably, the concentration decreases of Na and K were mainly ascribed to the adsorption of electrode, instead of the electrochemical extraction. After several washing processes by water, the majority of adsorbed Na and K could be removed from the electrode.

We have added the discussion on the concentration variation of metal species in the electrolyte involved in the electrochemical reaction in the revised manuscript (Page 15, lines 344-351).

Fig. R6 (a) and (b) The metal species concentrations of real wastewater and transparent residual liquid. (c) The proportion of metal species of transparent residual liquid.

Revision: Fig. R6 is added as Supplementary Fig. 20 in the revised Supplementary Information. Table R2. The ion concentration of metal species of transparent residual liquid after electrochemical uranium extraction.

Ion species	Concentration (mg L ⁻¹)	Proportion (%)
K	140939.8	95.908828
Na	6008.6	4.0888222
U	2.2	0.0014971
Mn	0.14	0.0000953
Al	0.023	0.0000157
Fe	1.00	0.0006805
Ca	0.09	0.0000612

Revision: Table R2 is added as Supplementary Table 4 in the revised Supplementary Information.

7. Authors should provide detailed XPS interpretation for Ti 2p of OH-terminated Ti_3C_2 nanosheets with respect to that of $\text{Ti}(\text{OH})\text{PO}_4$. It is worth mentioning the oxidation states of Ti atoms (Ti , Ti^{2+} , Ti^{3+}).

Reply: We appreciate the reviewer for providing these constructive comments to improve the quality of the article. Based on your suggestion, we have described the detailed XPS interpretation for Ti 2p of OH-terminated Ti_3C_2 and $\text{Ti}(\text{OH})\text{PO}_4$ for a better understanding (Fig. R7). As shown in Fig. R7, Ti_3C_2 possessed the dominant signal of the Ti-C bond, together with the slight signal of $\text{Ti}^{4+}\text{-O}$ and $\text{Ti}^{2+}\text{-O}$ bonds^{1,2,3}, which was attributed to the -OH termination. By comparison, the Ti 2p spectrum of $\text{Ti}(\text{OH})\text{PO}_4$ showed three sets of peaks, which were assigned as $\text{Ti}^{4+}\text{-O}$, $\text{Ti}^{3+}\text{-O}$, and $\text{Ti}^{2+}\text{-O}$. The massive low valent $\text{Ti}^{\delta+}$ ($\delta < 4$) indicated the defective nature of $\text{Ti}(\text{OH})\text{PO}_4$, which was constructed by the oxygen vacancies during the transformation of OH-terminated Ti_3C_2 into $\text{Ti}(\text{OH})\text{PO}_4$.

We have added the discussion on the oxidation state of the Ti atom in the revised manuscript (Page 7, lines 133-139).

Fig. R7 Ti 2p XPS spectrum, of $\text{Ti}(\text{OH})\text{PO}_4$ nanorods and OH-terminated Ti_3C_2 nanosheets

Revision: Fig. R7 is added as a new Fig. 2c in the revised manuscript.

References

1. Li, Y., Ding, L., Liang, Z., Xue, Y., Cui, H., Tian, J. Synergetic Effect of Defects Rich MoS_2 and Ti_3C_2 MXene as Cocatalysts for Enhanced Photocatalytic H_2 Production Activity of TiO_2 . *Chem. Eng. J.* **383**, 123178 (2020).

2. Li, R., Ma, X., Li, J., Cao, J., Gao, H., Li, T., Zhang, X., Wang, L., Zhang, Q., Wang, G., Hou, C., Li, Y., Palacios, T., Lin, Y., Wang, H., Ling, X. Flexible and High-Performance Electrochromic Devices Enabled by Self-Assembled 2D TiO₂/MXene Heterostructures. *Nat. Commun.* **12**, 1587 (2021).
3. Zhang, H., Yang, L., Zhang, P., Lu, C., Sha, D., Yan, B., He, W., Zhou, M., Zhang, W., Pan, L., Sun, Z. MXene-Derived Ti_nO_{2n-1} Quantum Dots Distributed on Porous Carbon Nanosheets for Stable and Long-Life Li-S Batteries: Enhanced Polysulfide Mediation via Defect Engineering. *Adv. Mater.* **33**, 2008447 (2021).

8. The authors should provide the experimental conditions of U adsorption through non-voltage adsorption method.

Reply: We appreciate the reviewer for providing these constructive comments to improve the quality of the article. Based on your suggestion, we have described the full experimental conditions of uranium adsorption through non-voltage adsorption in the Methods part of the revised manuscript (Revised manuscript Page 20, lines 468-472). The specific modifications are as follows:

The non-voltage adsorption experiments were tested on an unpowered three-electrode system in 50 mL of simulating nuclear water. The as-prepared Ti(OH)PO₄ electrode, Pt wire, and Ag/AgCl electrode were also used as the working electrode, counter electrode, and reference electrode respectively. The condition of the adsorption test was similar to that of electrochemical extraction except for excluding external current or voltage.

9. What would be the approximate operating cost of your proposed method to extract per kg of Uranium.

Reply: We thank the reviewer for this insightful suggestion. In 400 mL of real radioactive wastewater, we extracted 273 mg of U at the constant current density of 30 mA/cm² for 7 h using 25 cm² of the electrode. The operated voltage of the two-electrode system was ~6 V, thereby the consumed electricity was 31.5 W h. This result means the extraction per kg of U requires electricity of ~115 kWh, corresponding to an electricity cost of ~8 US dollars (according to the price of China). For the cost of Ti(OH)PO₄ electrode, the price was approximately 0.2 US dollars considering the purchase of Ti₃AlC₂ precursor, synthetic material, and graphite support. Considering the recycling use of the electrode, the cost of electrode material was ~14 US dollars

for extracting 1 kg of uranium. Therefore, the overall cost was approximately 22 US dollars for the extraction of 1 kg U considering both energy consumption and material costs.

We have added the discussion on the approximate operating cost in the revised manuscript (Page 17, lines 374-381).

Reviewer #2

This manuscript reported on the application of electrochemical uranium extraction from nuclear wastewater recycling uranium resources. It constructs $\text{Ti}^{\delta+}\text{-PO}_4^{3-}$ ion pair extraction sites in $\text{Ti}(\text{OH})\text{PO}_4$ for efficient electrochemical uranium extraction in wastewater from nuclear fuel production. These sites can selectively bind with UO_2F_x through the combined Ti-F bond and multiple O-U-O bonds. In real nuclear wastewater, the electrochemical extraction efficiency of uranium reaches 99.6% within 7 h, corresponding to a remarkable extraction capacity of 6829 mg g^{-1} without saturation. However, some aspects of the current manuscript should be improved. Thus, I recommend major revision before its publication.

Reply: We greatly appreciate your professional comments on our article. As you are concerned, several issues need to be resolved. Based on your suggestion, we have made extensive corrections to the previous manuscript, and the specific corrections are as follows.

1. In Fig. 2, the Ti_3C_2 needs to be characterized by Fourier transform infrared to prove the change of its surface functional groups.

Reply: We appreciate the reviewer for providing these constructive comments to improve the quality of the article. According to the reviewer's suggestion, the changes in surface functional groups of Ti_3C_2 during the preparation process were elucidated by FTIR (Fig. R8). As shown in Fig. R8, characteristic peaks of accordion-like stacked Ti_3C_2 nanosheets were located at 3450 cm^{-1} , 1630 cm^{-1} , 1080 cm^{-1} , and 572 cm^{-1} , which corresponded to the stretching of $-\text{OH}$, $\text{C}=\text{O}$, $\text{C}-\text{F}$, and $\text{Ti}-\text{O}$ groups, respectively^{1,2}. By comparison, the peak at 1080 cm^{-1} disappeared for exfoliated lamellar OH-terminated Ti_3C_2 nanosheets, demonstrating the replacement of $-\text{F}$ terminations by $-\text{OH}$ terminations during the exfoliation process.

We have added the discussion on the FTIR characterization in the revised manuscript (Page 4, lines 85-89).

References

1. Xue, Q., Zhang, H., Zhu, M., Pei, Z., Li, H., Wang, Z., Huang, Y., Huang, Y., Deng, Q., Zhou, J., Du, S., Huang, Q. & Zhi, C. Photoluminescent Ti_3C_2 MXene quantum dots for multicolor cellular imaging. *Adv. Mater.* **29**, 1604847 (2017).
2. Zhou, K., Gong, K., Gao, F. & Yin, L. Facile strategy to synthesize MXene@LDH nanohybrids for boosting the flame retardancy and smoke suppression properties of epoxy. *Compos. Part A Appl. Sci. Manuf.* **157**, 106912 (2022).

Fig. R8 FTIR spectra of as-synthesized accordion-like stacked Ti_3C_2 nanosheets and exfoliated lamellar OH-terminated Ti_3C_2 nanosheets.

Revision: Fig. R8 is added as Supplementary Fig. 3 in the revised Supplementary Information.

2. When studying the influence of other ions and pH, it is necessary to detect the changes in ion concentration and pH before and after the reaction.

Reply: We appreciate the reviewer for providing these constructive comments to improve the quality of the article. According to the reviewer's suggestion, we have tested the changes in pH and ion concentrations before and after the electrochemical uranium extraction reaction (Fig. R9). As shown in Fig. R9a, regardless of the wide manipulation of pH from 4 to 10, all the pH values of the electrolyte increased after the electrochemical reaction because of the hydrogen evolution in the electrochemical process. Moreover, the concentrations of anions in the electrolyte displayed a negligible decrease after the electrochemical reaction, which was attributed to the slight adsorption of ions by the electrode (Fig. R9b).

Additionally, we also measured the concentration of metal species involved in the real radioactive wastewater experiment by ICP-OES and ICP-MS. As shown in Fig. R10, the concentration of K and Na decreased by 0.5% and 2.2% after the reaction, which was similarly ascribed to the adsorption of the electrode, instead of the electrochemical extraction. After several washing processes by water, the majority of adsorbed Na and K could be removed from the electrode. For other trace metal elements, the concentrations also exhibited a decrease, whereas their low initial concentrations (below 2 mg L⁻¹) were essentially inconsequential to the uranium extraction process and purity of uranium deposits.

We have added the discussion on ion concentration and pH changes in the revised manuscript (Page 9, lines 204-210, and Page 10, line 211).

Fig. R9 (a) pH value changes and (b) the concentration of other ions changes before and after reaction.

Revision: Fig. R9 is added as Supplementary Fig. 11 in the revised Supplementary Information.

Fig. R10 (a) and (b) The metal species concentrations of real wastewater and transparent residual liquid.

Revision: Fig. R10 is added as Supplementary Fig. 20 in the revised Supplementary Information.

3. The author emphasized the influence of F^- on uranium recovery in the front, but did not explain in detail the change and role of F^- in uranium recovery in this study. The author should supplement the influence of F^- concentration change on uranium recovery and the change of F^- ion concentration before and after recovery.

Reply: We thank the reviewer for this insightful suggestion. Based on your comment, we have supplemented the influence of F^- concentration change on uranium recovery and the change of F^- concentrations before and after recovery (Fig. R11). Considering the practical situation of nuclear production, the initial concentration of F^- varied from 5 g L^{-1} to 30 g L^{-1} , which represented the real wastewater between the valley point and the peak point. As shown in Fig. R11a, regardless of the initial concentration of F^- , the U(VI) concentration was reduced from approximately 100 mg L^{-1} to $\sim 1 \text{ mg L}^{-1}$ after the reaction with a high level of $>98\%$ (Fig. R11b). This result indicated that the F^- concentrations did not exhibit an obvious influence on the U(VI) extraction for the $Ti(OH)PO_4$ electrode with ion pair sites, which was attributed to the selective binding of UO_2F_x , instead of UO_2^{2+} .

Furthermore, slight decreases in the concentration of F^- were observed after the electrochemical reaction (Fig. R11c). This decrease was consistent with the evolution phenomenon of uranium species. The $Ti^{\delta+}$ cation site was able to adsorb F^- ions, which participated in the transformation of U_3O_7 into $K_3UO_2F_5$. Notably, due to the ultrahigh molar ratio of F/U (ranging from 626 to 3758) in the electrolyte, the F^- ions consumed by the crystallization of $K_3UO_2F_5$ were negligible.

We have added the discussion on the influence of F^- concentration change on uranium recovery and the change of F^- ion concentration in the revised manuscript (Page 10, lines 211-219).

Fig. R11 (a) The change in uranium concentration after electrochemical uranium extraction under the presence of electrolytes with different F^- concentrations (ranging from 5 g L^{-1} to 30 g L^{-1})

L⁻¹). (b) The electrochemical extraction efficiency of U(VI) on Ti(OH)PO₄ under different concentrations of F⁻ varied from 5 g L⁻¹ to 30 g L⁻¹. (c) The change of F⁻ concentrations before and after the reaction.

Revision: Fig. R11 is added as new Fig. 3e and Supplementary Fig. 12 in the revised manuscript.

4. The author should provide more direct proof for the change of uranium valence state.

Reply: We appreciate the reviewer for providing these constructive comments to improve the quality of the article. To validate the change in the uranium valence state, we have supplemented the U 4f XPS spectrum of U-Ti(OH)PO₄ electrodes at different time points during the electrochemical extraction (Fig. R12). At the time point of 1 h, the deposited species appeared and exhibited a grey color. The peak of the U 4f_{7/2} was observed at 381.5 eV, which was a special species of UO_{2+x} containing U(V) and U(IV). With the reaction proceeding to 3 h, the peaks of U 4f_{7/2} were negatively shifted to 380.7 eV, indicating the continuous reduction of uranium during the reaction. With the reaction proceeding to 4 h and 7 h, the grey deposit was gradually transformed into the yellow product, together with the aggregation of the solid deposits on the electrode. At time point of 7 h, the peaks of U 4f_{7/2} were positively shifted to 381.8 eV, indicating the oxidation of low valent uranium in the deposits during the reaction. In general, the UO₂F_x was firstly reduced to low-valent species, resulting in the formation of an intermediate UO_{2+x} product. The metastable low-valent uranium was then oxidized, followed by crystallizing with F⁻ and forming the final K₃UO₂F₅.

Fig. R12 U4f XPS spectrum of U-Ti(OH)PO₄ electrodes at different electrochemical extraction

time points.

Revision: Fig. R12 is added as a new Fig. 4c in the revised manuscript.

We have added the discussion on the change of uranium valence state in the revised manuscript (Page 11, lines 253-254, and Page 12, lines 255-258).

5. The composition of electrolyte should be indicated in the experimental method, whether the K element in $K_3UO_2F_5$ comes from electrolyte, whether the change of electrolyte will change the product, and whether the K and F in the product can be further removed.

Reply: We thank the reviewer for this insightful suggestion. Based on your comment, we have supplemented the composition of the electrolyte in the experimental method. The specific modifications are as follows: The electrochemical uranium extraction performances were tested in an aqueous solution (50 mL) containing 100 mg L^{-1} of U(VI) and 30 g L^{-1} of F^- (F^- were sourced from KF electrolyte). The K in the final uranium extraction product ($K_3UO_2F_5$) originated from the KF in the electrolyte.

Revision: We have added the discussion on the composition of electrolytes in the revised manuscript (Page 20, lines 466-468).

Fig. R13 (a) XRD pattern of the collected solid deposits at 30 g L^{-1} NaF electrolyte. (b) XRD pattern of the collected solid deposits at 15 g L^{-1} NaF and 15 g L^{-1} KF electrolyte. (c) XRD pattern of the collected solid deposits at 30 g L^{-1} KF electrolyte.

Revision: Fig. R13 is added as Supplementary Fig. 13 in the revised Supplementary Information.

We also investigated the variations in uranium extraction products on the $Ti(OH)PO_4$ electrode in uranium-containing electrolytes of different compositions (pure NaF, hybrid NaF and KF with mole ratio of 1:1). As indicated by the XRD pattern (Fig. R13a), the extraction product of uranium was identified as $Na_3UO_2F_5$ (JCPDS #33-1300) after 7-h electrolysis at a

constant current density of 30 mA cm^{-2} in pure NaF electrolyte. By comparison, the XRD of deposits in hybrid NaF and KF hybrid electrolytes fitted by $\text{Na}_3\text{UO}_2\text{F}_5$ and $\text{K}_3\text{UO}_2\text{F}_5$ (JCPDS #38-0023) (Fig. R13b). In the pure KF electrolyte, the uranium extraction product was $\text{K}_3\text{UO}_2\text{F}_5$ which was mentioned in the manuscript. Consequently, the species of uranium crystallization product was directly affected by the type of metal ions present in high concentrations within the electrolyte (Fig. R13c).

We have added the discussion on the composition of electrolytes on uranium products in the revised manuscript (Page 10, line 241, and Page 11, lines 242-243).

In addition, we employed a typical precipitation-calcination process for the further purification of uranium products (Fig. R14a). Initially, we collected and dissolved the uranium extraction product in HCl. Subsequently, excess ammonia was added to the above solution to form a precipitate of ammonium diuranate. Finally, ammonium diuranate was collected by centrifugation and heated at $550 \text{ }^\circ\text{C}$ for 5 h to form a black-green uranium product. According to XRD and ICP-OES tests, the purified black-green uranium product was identified as U_3O_8 (JCPDS #31-1425) with a uranium content of 99.93% among the metal elements (Fig. R14b and R14c), which can be directly used in the uranium production.

We have added the discussion on the further purification for the movement of K and Na in the revised manuscript (Page 17, lines 365-372)

Fig. R14 (a) The schematic diagram of the precipitation-calcination purification process for uranium. (b) XRD pattern of the purification product. (c) The proportion of uranium among the metal species in the purification product.

Revision: Fig. R14 is added as new Fig. 6e and Supplementary Fig. 21 in the revised manuscript.

6. Please provide a more intuitive energy diagram of the splitting orbits of Ti 3d peak.

Reply: We thank the reviewer for raising this issue. Firstly, we feel sorry for the typo of wrong labels for the curves in Fig. 2d in the previous figure, which led to an unnecessary misunderstanding. In the revised manuscript, we have corrected the label and additionally provided a table (Table R3) accompanied by the energy diagram to indicate the intuitive splitting orbits of the Ti 3d peak and a revised energy diagram of the splitting orbits of Ti 3d peak (Fig. R15). Typically, Ti possessed five 3d orbits, which could be divided into two e_g orbits and three t_{2g} orbits. The binding energy of e_g and t_{2g} orbits was associated with two factors, including the coordination environment and the electron filling of Ti. For the coordination environment, the transformation of Ti-C in OH-terminated T_3C_2 into Ti-O in $Ti(OH)PO_4$ induced the positive shift of both e_g and t_{2g} orbits in Ti L-edge XANES, owing to the higher electronegativity of O than that of C. Moreover, for the factor of electron filling, Ti^{4+} possessed empty e_g and t_{2g} orbits, whereas defective $Ti(OH)PO_4$ possessed abundant Ti^{3+} and Ti^{2+} , which respectively had one and two electrons in 3d orbits. Due to the lower energy of t_{2g} orbits than e_g orbits, the additional electrons in $Ti^{\delta+}$ were located at t_{2g} orbits, resulting in the dramatic energy decrease of t_{2g} orbits. As a comprehensive result of the coordination environment and the electron filling of Ti, the e_g peak in $Ti(OH)PO_4$ showed an obvious shift, whereas t_{2g} peak displayed a negligible shift relative to OH-terminated T_3C_2 in Ti L-edge XANES (Table R3).

We have added the discussion on the energy diagram of the splitting orbits of the Ti 3d peak in the revised manuscript (Page 7, lines 145-156).

Table R3 The shift analysis of e_g and t_{2g} orbits in Ti L-edge XANES during the transformation of OH-terminated T_3C_2 into $Ti(OH)PO_4$.

Factor	e_g orbits	t_{2g} orbits
Coordination environment (Ti-C into Ti-O)	Positive shift	Positive shift
Electron filling (existence of $Ti^{\delta+}$)	No change	Negative shift
Comprehensive Result	Positive shift	Negligible shift

Revision: Table R3 is added as a Supplementary Table 1 in the revised Supplementary

Information.

Fig. R15 XANES spectrum of Ti(OH)PO₄ nanorods and lamellar OH-terminated Ti₃C₂ nanosheets, and corresponding energy diagram of the splitting orbits of Ti 3d peak.

Revision: Fig. R15 is added as new Fig. 2d and Fig. 2e in the revised manuscript.

REVIEWERS' COMMENTS

Reviewer #1 (Remarks to the Author):

Many thanks for your revised manuscript. I believe you have answered all of my previous comments well and, for me at least, made the paper significantly clearer and more relevant to the efficient electrochemical extraction of uranium.

Therefore, I would recommend the publication as it stands.

Reviewer #2 (Remarks to the Author):

The manuscript can be accepted now.

Point-by-point response to reviewer comments

Manuscript ID: NCOMMS-23-60509B

MS Type: Article

Title: "Ion pair sites for efficient electrochemical extraction of uranium in real nuclear wastewater"

First of all, we sincerely thank the editor and all Reviewers for giving us valuable and thoughtful comments to improve the quality of this manuscript.

Reviewer #1

Many thanks for your revised manuscript. I believe you have answered all of my previous comments well and, for me at least, made the paper significantly clearer and more relevant to the efficient electrochemical extraction of uranium.

Therefore, I would recommend the publication as it stands.

Reply: We thank this reviewer for the support of publication.

Reviewer #2

The manuscript can be accepted now.

Reply: We thank this reviewer for the support of publication.